# Flexible active-matrix micro-LED display with 1T-1FeMFET architecture featuring scaling-limit-free design

Tingrui Huang ®[1,6], Guangan Yang[2,6], Yimeng Sang[3,6], Fulin Zhuo[4], Mingming Liu[1], Gengyu Li[1], Heng Cheng[1], Zuoxu Yu[1], Zhiyuan Fu[1], Siyang Liu[1], Haoliang Shen[5], Zhihao Yu ®[2], Bin Liu ®[4], Xinran Wang ®[4,5], Wangran Wu ®[1] ✉, Runxiao Shi ®[1] ✉, Zhe Zhuang ®[3] ✉ & Weifeng Sun ®[1] ✉

The advancement of flexible electronics necessitates displays that combine bendability, high resolution, and energy efficiency. Nevertheless, conventional pixel architectures impose critical limitations in power consumption and scaling, hindering the development of such displays. Here, we demonstrate a flexible active-matrix micro light-emitting diode display using a ferroelectric metal field-effect transistor with hafnium-based gate stack and an indium tin oxide channel, functioning as driver and memory element, fabricated below 400 °C on polyimide. The 400 °C-activated ferroelectric capacitors in transistors exhibit a remnant polarization of 47 $\mu C/cm^2$. The resulting devices achieve a record normalized memory window of 0.63 V/nm (7.5 V), an on/off ratio of $4 \times 10^8$, and robust flexibility, retaining performance after $10^5$ bending cycles at a radius of 4 mm. The proposed pixel circuit supports dual-mode driving schemes, enabling precise grayscale control at a 200 kHz refresh rate. This pixel architecture achieves a high resolution of 428 pixels per inch and dynamic power consumption of 0.68 nW, highlighting its potential for next-generation wearable and portable displays.

Flexible displays have garnered considerable interest for applications in electronic paper, dynamic signage, and next-generation portable electronics, owing to their inherent bendability, light weight, and wearability[1–5]. As market demands evolve toward more sophisticated devices, high pixel density and ultra-low power consumption have emerged as essential benchmarks for advanced flexible display systems. Conventional driving architectures, such as multi-transistor and capacitor (mT-nC) designs like 2T-1C and 7T-1C rely on large storage capacitors to maintain driving voltage stability and luminance uniformity (Fig. 1a). However, this approach intrinsically limits pixel miniaturization and restricts further enhancements in resolution.

Moreover, the need for periodic capacitor refreshes results in substantial dynamic power consumption, which contradicts the stringent energy efficiency demands of modern portable and wearable devices[6–10]. Thus, prevailing mT-nC schemes represent a major bottleneck in realizing high-resolution, low-power flexible displays, underscoring the critical need for other driving methodologies that reconcile capacitor size, scaling capability, and power efficiency.

The $Hf_{1-x}Z_xO_2$ (HZO) -based ferroelectric capacitors (FeCAPs) and ferroelectric field-effect transistors (FeFETs), which exhibit non-volatile ferroelectric polarization states, have attracted huge interests due to advantages such as nondestructive readout, fast program/

[1]School of Integrated Circuits, National ASIC System Engineering Research Center, Southeast University, Nanjing, China. [2]College of Integrated Circuit Science and Engineering, Nanjing University of Posts and Telecommunications, Nanjing, China. [3]School of Integrated Circuits, Nanjing University, Nanjing, China. [4]School of Electronic Science and Engineering, Nanjing University, Nanjing, China. [5]Suzhou Laboratory, Suzhou, China. [6]These authors contributed equally: Tingrui Huang, Guangan Yang, Yimeng Sang. ✉e-mail: wrwu@seu.edu.cn; icrshi@seu.edu.cn; zzhuang@nju.edu.cn; swffrog@seu.edu.cn

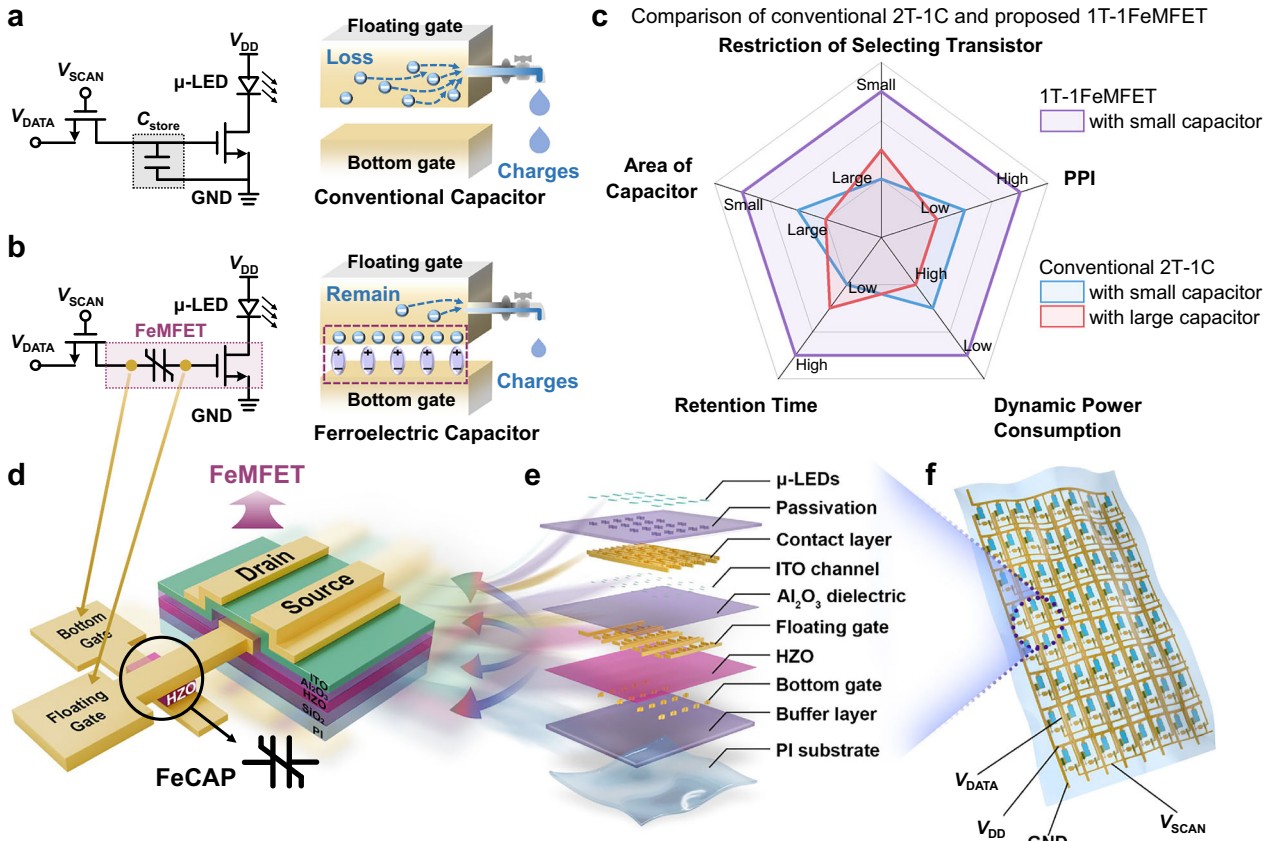

**Fig. 1 | Schematic and advantages of 1T-1FeMFET pixel circuit.** Schematic diagram of conventional 2T-1C pixel circuit (**a**) and the proposed 1T-1FeMFET pixel circuit (**b**). Benefited from the non-volatile property of ferroelectric capacitors, the charges stored in FeMFET exhibit remarkable retention with minimal loss. **c** Comparisons of conventional 2T-1C and proposed 1T-1FeMFET. The 1T-1FeMFET

with small size FeCAP exhibits great promising for long retention time, high PPI, and low dynamic power consumption. **d** Device structure diagram of the ITO FeMFET with an adjustable FeCAP. Schematic view of the detailed cross-sectional layer structures (**e**) and the fabricated flexible active-matrix μ-LED displays (**f**).

read-out speed, and CMOS compatibility, making them a promising platform for the non-volatile memory (NVM) applications[11–21]. Building on these features, the ferroelectric metal field-effect transistor (FeMFET) architecture, which incorporates a metal layer between the ferroelectric and dielectric interfaces, has been shown to markedly reduce depolarization effects and improve switching stability, making it especially suitable for machine learning and in-memory computing[22,23]. However, reports on excellent ferroelectric properties in HZO and the subsequent high-performance FeMFETs on flexible substrates remains scarce, compounded by a lack of systematic studies on their electrical and mechanical properties. These limitations pose a key constraint on the development of FeMFETs for flexible electronics[24–28]. Within oxide semiconductors, where p-type materials are also advancing[29,30], indium tin oxide (ITO) stands out as a nanoscale channel material offering a wide bandgap (3.5–4.2 eV), competitive mobility around 30 cm²V⁻¹s⁻¹, and ultra-low off-state current. These characteristics are achieved through low-temperature (<200 °C) deposition techniques compatible with polyimide (PI) substrates and back-end integration processes[31–35]. Its suitability for large-area fabrication and practical mechanical properties further makes ITO a compelling channel choice for FeMFETs[36,37].

In this study, we present an innovative integration of FeMFET and ITO channel, culminating in the design and fabrication of a 1T-1FeMFET pixel circuit. By leveraging the non-volatile characteristics of ferroelectric HZO alongside the high on/off ratio of ITO, the proposed 1T-1FeMFET pixel circuit features a scaling-limit-free structure with high resolution and low dynamic power consumption, overcoming the

inherent limitations associated with traditional 2T-1C pixel circuits. We report a high-performance flexible active-matrix (AM) micro light-emitting diode (μ-LED) display driven by 1T-1FeMFET pixel circuits integrated on a PI substrate with processing temperature below 400 °C. The 400 °C-activated HZO FeCAPs achieve a remnant polarization (2$P_r$) of 47 μC/cm². Meanwhile, the ITO channel-based FeMFETs exhibit a record memory window of 0.63 V/nm and a high on/off ratio of 4 × 10⁸. The pixel circuit supports dual-mode driving schemes (PAM/PWM) for accurate grayscale control and operates at a refresh rate of 200 kHz with ultra-low dynamic power consumption of 0.6 nW. The small FeCAP (9 μm²) enables significant pixel shrinkage, yielding a resolution of 428 pixels per inch (PPI). Robust mechanical flexibility is confirmed through repeated bending tests down to a 4 mm radius. Finally, we successfully integrate and address GaN-based μ-LED arrays via a transfer and bonding process, validating the feasibility of this architecture for future high-resolution, energy-efficient flexible displays.

## Results

### Superiority of the proposed 1T-1FeMFET pixel circuit
Due to the inevitable loss of charges in the storage capacitor, conventional 2T-1C pixel circuits reply on large storage capacitors and optimized selecting transistors to maintain driving voltage stability and luminance uniformity (Fig. 1a). However, this approach intrinsically limits pixel miniaturization and restricts further enhancements in resolution. Moreover, the need for periodic capacitor refreshes results in substantial dynamic power consumption, which contradicts the

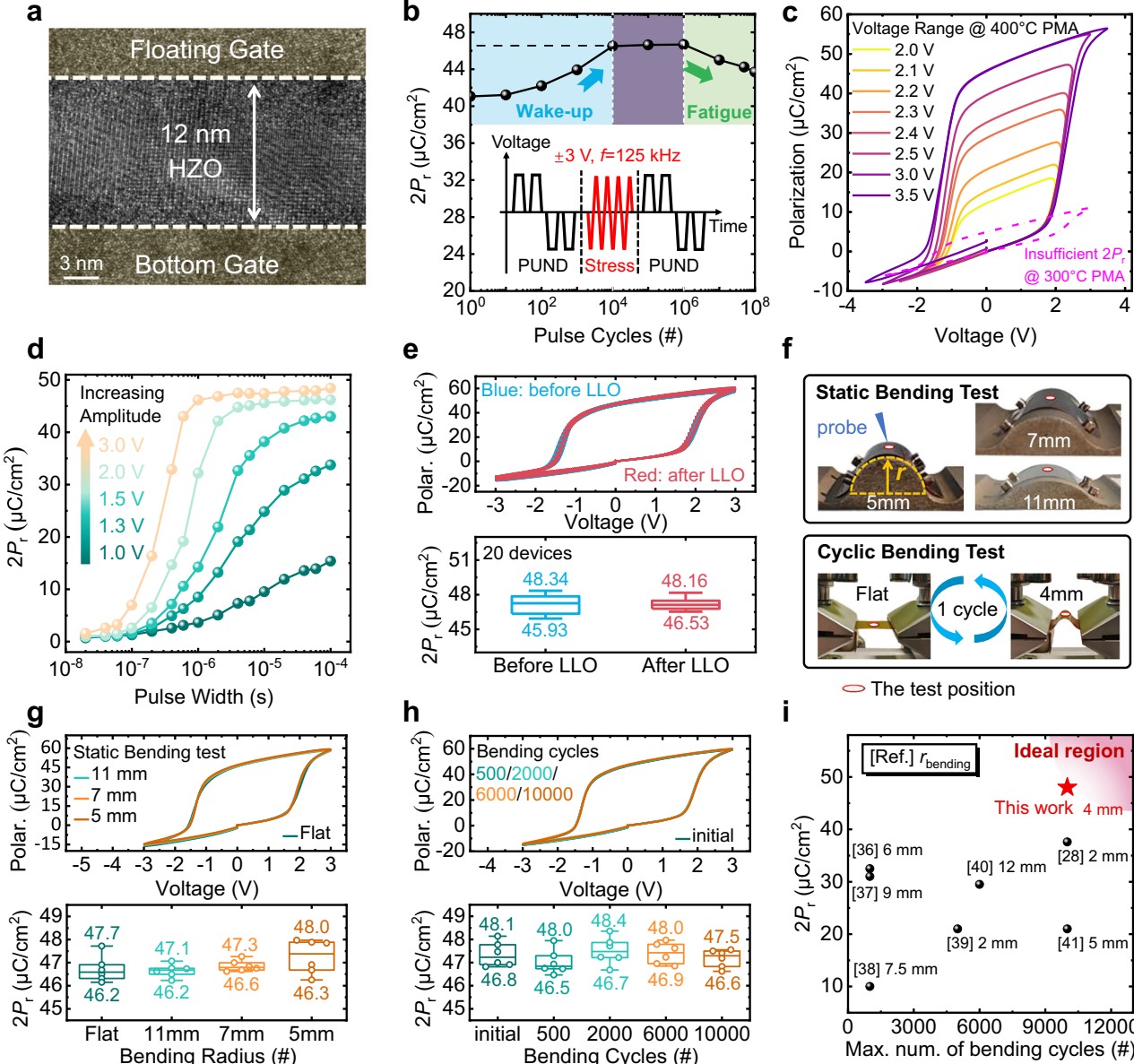

**Fig. 2 | Characterization and performance of flexible FeCAP. a** TEM image of the 12-nm thick HZO layer, which is embedded between top and bottom Mo electrodes. High crystallinity of HZO is well defined. **b** Endurance properties of the FeCAP. Inset shows the schematic of measurement set-up for endurance. Endurance over $1 \times 10^8$ cycles is accomplished under cycle stress at voltage amplitude of 3 V and frequency of 125 kHz. **c** P-V hysteresis loop at various voltage ranges. **d** The polarization as a function of programming pulse amplitudes and pulse widths of the FeCAP. **e** P-V hysteresis loop and extracted $2P_r$ of 20 FeCAPs before and after laser-lifted-off (LLO). **f** Photograph of samples on PI film under static and cyclic bending tests, with various bending radius ($r_{bending} = 11$ mm, 7 mm and 5 mm) and bending cycles up to

$10^5$. Samples were positioned within the red-circled regions to ensure they were subjected to the intended and consistent stress. P-V hysteresis loops and extracted $2P_r$ of FeCAPs at various $r_{bending}$ (**g**) and after various bending cycles (**h**). Box plot elements of this work: center line, median; box limits, first quartile and third quartile; whiskers, 1.5×interquartile range. The FeCAPs show no degradation. **i** Benchmarking of $2P_r$ versus maximum number of bending cycles of the flexible FeCAP reported in this work (red star) and recently reported HZO-based FeCAPs (black balls). The 400 °C-activated FeCAP attains the highest $2P_r$ of 47 μC/cm², while enduring $10^5$ bending cycles at 4 mm radius without any degradation.

stringent energy efficiency demands of modern portable and wearable devices. In contrast, a strategy—the proposed 1T-1FeMFET pixel circuit —is introduced to address the aforementioned issues and limitations (Fig. 1b). By harnessing the remnant polarization of a ferroelectric gate stack layer, the ITO FeMFET intrinsically merges the functions of a driving transistor and an embedded memory element, eliminating the need for a separate storage capacitor and enabling substantial pixel shrinkage. As illustrated in Fig. 1c, the proposed 1T-1FeMFET pixel circuit with exceptional scaling potential effectively addresses the trade-off between charge storage and capacitance area when

compared to the conventional 2T-1C pixel circuit. This advancement demonstrates superior dynamic power consumption and display resolution, coupled with remarkable retention time and the elimination of restrictions on selecting transistors.

## Ferroelectricity and flexibility properties of FeCAPs

Figure 1d shows schematic view of the FeMFET, with the HZO-based FeCAP component highlighted by a red circle. The transmission electron microscope (TEM) image of HZO layer is shown in Fig. 2a. The high crystallinity of the 12-nm-thick HZO layer is well defined after the

post metallization annealing (PMA) treatment at 400°C. This temperature, along with a 12-nm HZO thickness, is found to optimize ferroelectric performance after systematic variation of these parameters (Supplementary Fig. 1). The process details are illustrated in Methods and Supplementary Fig. 18. Figure 2b shows the endurance properties of FeCAP. Inset depicts the schematic of measurement set-up for endurance. High endurance exceeding $1 \times 10^8$ cycles is accomplished under cycle stress with a voltage amplitude of 3 V (@ 0.25 MV/cm across the HZO layer) and frequency of 125 kHz. $2P_r$ increases from 41 μC/cm$^2$ to 47 μC/cm$^2$ during wake-up, and decreases to 44 μC/cm$^2$ during fatigue, finally breakdowns after $10^8$ cycles. Supplementary Fig. 2 presents the detailed P-V hysteresis loops measured during the wake-up process, showcasing a 15% increase in $2P_r$ over $10^5$ cycles. All subsequent P-V hysteresis measurements and $2P_r$ values were obtained from FeCAPs following the wake-up process. As shown in Fig. 2c, the ferroelectric polarization undergoes complete switching at voltages above 3 V, although this yields insufficient $2P_r$ for FeCAPs after 300 °C PMA. Reducing the sweep amplitude enables partial polarization switching (Supplementary Fig. 3). Figure 2d compares the difference in ferroelectric switching polarization under varying pulse widths and amplitudes, obtained using an initialize-modulate-measure-verify sequence (Supplementary Fig. 4). Nearly complete switching is achieved with 3 V, 600 ns pulses.

Supplementary Fig. 5a shows an optical image of the PI film after laser lift-off (LLO). Owing to its low thermal shrinkage (<0.1% at 400 °C) and coefficient of thermal expansion, the released PI film exhibits minimal residual stress and remains flat without a rigid carrier. Figure 2e compares the P-V hysteresis loops and extracted $2P_r$ values of 20 FeCAPs before and after LLO. The loops show no significant distortion or shift after release, and the $2P_r$ values retain high uniformity across devices, indicating minimal impact from the LLO process. Figure 2f presents optical images of samples on a PI film during static and cyclic bending tests, enabling a systematic evaluation of flexibility. For static bending test, samples are mounted on arched fixtures with bending radius ($r_{bending}$) of 11 mm, 7 mm, and 5 mm, which correspond to calculated bending strains ($ε$) of 0.17%, 0.27%, and 0.38%, respectively. Cyclic bending tests defined one bending cycle as a transition from flat to 4-mm $r_{bending}$ ($ε$ of 0.48%) and back, repeated for $10^5$ cycles. Test areas are positioned at the red circle region to confirm the target stress condition consistent. Figure 2g, h present the P-V hysteresis loops and corresponding extracted $2P_r$ values of the FeCAPs under various bending radius and after repeated bending cycles. The electrical characteristics remain stable across all tested conditions, including $10^5$ cycles at 4 mm radius, demonstrating good mechanical durability and operational reliability.

Figure 2i benchmarks the relationship between $2P_r$ and maximum number of bending cycles of flexible FeCAPs (red star) in this work and recently reported HZO-based FeCAPs (black balls)[28,38–43]. Simultaneously achieving high polarization endurance and mechanical flexibility in FeCAPs remains challenging due to the thermal constraints of polymer substrates and the high crystallization temperature required for HZO (ideal region marked in pink). The fabricated FeCAP attains the highest $2P_r$ of 47 μC/cm$^2$−25% higher than prior reports−while withstanding $10^5$ bending cycles at 4 mm radius without any degradation. These results indicate robust operational stability under repeated mechanical stress for the flexible FeCAP (see Supplementary Table 1 for details).

### Device performance of flexible ITO FeMFET

As shown in Fig. 1d, the ITO FeMFET structure incorporates both bottom gate (BG) and floating gate (FG) electrodes, supporting operation in two distinct modes (Supplementary Fig. 6). When a voltage is applied to the FG ($V_{FG}$), the device operates in dielectric mode (De-mode), effectively screening the polarization effect of the HZO layer and functioning as a conventional ITO TFT. Conversely, applying

voltage to the BG ($V_{BG}$) activates ferroelectric mode (Fe-mode), where the HZO polarization directly modulates the conductivity and carrier density of the ITO channel. Figure 3a presents the transfer at $V_d = 0.1$ V and output at $V_{FG} = 2$ to 5 V characteristics of an ITO FeMFET with $W/L = 50$ μm/15 μm and a FeCAP size of $3 \times 3$ μm$^2$. The device achieves a field-effect mobility ($μ_{FE}$) of 30 cm$^2$V$^{-1}$s$^{-1}$, a threshold voltage ($V_{th}$) of 2.2 V (defined as the gate voltage producing a normalized drain current of $W/L \times 100$ pA), and an off-state current of $2 \times 10^{-10}$ μA/μm. Due to ambient-induced parasitic conduction path at the back channel, a minor hump is observable[44–49]. The ultra-low off current arises from the wide bandgap and low intrinsic carrier concentration of the ITO semiconductor. These characteristics collectively result in a high on/off ratio of $4 \times 10^8$. Notably, the measured off-state current corresponds to the ~ 10 fA detection limit of the measurement setup (see Supplementary Fig. 7), implying an even lower intrinsic value.

Figure 3b illustrates the hysteresis transfer characteristics of the ITO FeMFET across different ranges of $V_{BG}$, displaying clear counter-clockwise hysteresis loops that expand with increasing $V_{BG}$ sweep amplitude. The $V_{th}$ differences from the forward ($V_{th-pro.}$) and backward ($V_{th-era.}$) sweeps define the memory window (M.W.), which reaches 7.5 V at $V_{BG}$ range of ± 8 V, as depicted in Fig. 3c. It leads to a high normalized M.W. (M.W. divided by HZO thickness) of 0.63 V/nm. Figure 3d benchmarks the normalized M.W. against on/off ratio, comparing this work (red star) with previously reported HZO-based FeFETs on flexible (yellow triangles) and rigid (blue squares) substrates[13,22,50–56]. The ITO FeMFET realizes a high M.W. of 7.5 V, leading to a record normalized M.W. of 0.63 V/nm, and an excellent on/off ratio of $4 \times 10^8$ (Supplementary Table 2 for details).

Superior on/off ratio derives from good carrier mobility and intrinsic ultra-low off current of ITO channel. The record M.W. of 0.63 V/nm (7.5 V absolute) stems from the synergistic effect between ferroelectric (FE) switching in the HZO layer and charge trapping at the floating-gate (FG) node. With a channel dimension ($A_{De}$) of 750 μm$^2$ and a FeCAP area ($A_{Fe}$) of 9 μm$^2$, the resulting high area ratio (AR = $A_{De}/A_{Fe}$) of 83 causes the majority of $V_{BG}$ to drop across the HZO layer, thus enabling complete polarization reversal. This FE-switching yields a fundamental M.W. of approximately 3 V, which approaches twice the coercive voltage ($2V_C$) of the HZO layer. Beyond this, the considerable thickness (20 nm) of the Al$_2$O$_3$ layer, along with a relatively small voltage dropped across it, results in the charge transfer effect between the BG and FG overwhelming that between the ITO and FG (Supplementary Fig. 8). Subsequently, charge trapping further expands the M.W. beyond the conventional $2V_C$ limit to 7.5 V[57–61]. Therefore, these two processes act synergistically to achieve the recorded M.W.; Besides, the hysteresis transfer curves of the ITO FeMFETs with a fixed $A_{De}$ of 750 μm$^2$ while varying $A_{Fe}$ are investigated (Supplementary Fig. 9), which also serves to validate the aforementioned synergistic effect. To further explore the geometry dependence, with a focus on the channel-reduction trend, devices with a fixed $A_{Fe}$ (9 μm$^2$) but scaled $A_{De}$ are characterized (Fig. 3e and Supplementary Fig. 10). The M.W. narrows with decreasing $A_{De}$, dropping from 9.4 V at 1000 μm$^2$ to 1.2 V at 150 μm$^2$, where a weakened counterclockwise hysteresis remains. Further scaling to 25 μm$^2$ eliminates the ferroelectric signature, resulting in a purely clockwise hysteresis.

Supplementary Fig. 11 defines the specific currents used in static and transient measurements. In static hysteresis curves (Supplementary Fig. 11a), $I_{on}$ ($I_{off}$) is defined as $I_d$ at $V_{BG} = 0$ V during the backward (forward) sweep, corresponding to point B (D). Similarly, in transient pulse tests (Supplementary Fig. 11b), $I_{on}$ ($I_{off}$) is measured at $V_{BG} = 0$ V after applying a positive (negative) programming pulse. Supplementary Fig. 12 further shows the output characteristics after programming and erasing pulses, demonstrating effective modulation of the device's on-resistance ($R_{on}$) through ferroelectric polarization switching. Figure 3f presents the dependence of $I_{on}$ (at $V_d = 0.1$ V) on programming pulse amplitude and width in the ITO FeMFET. Systematic control of

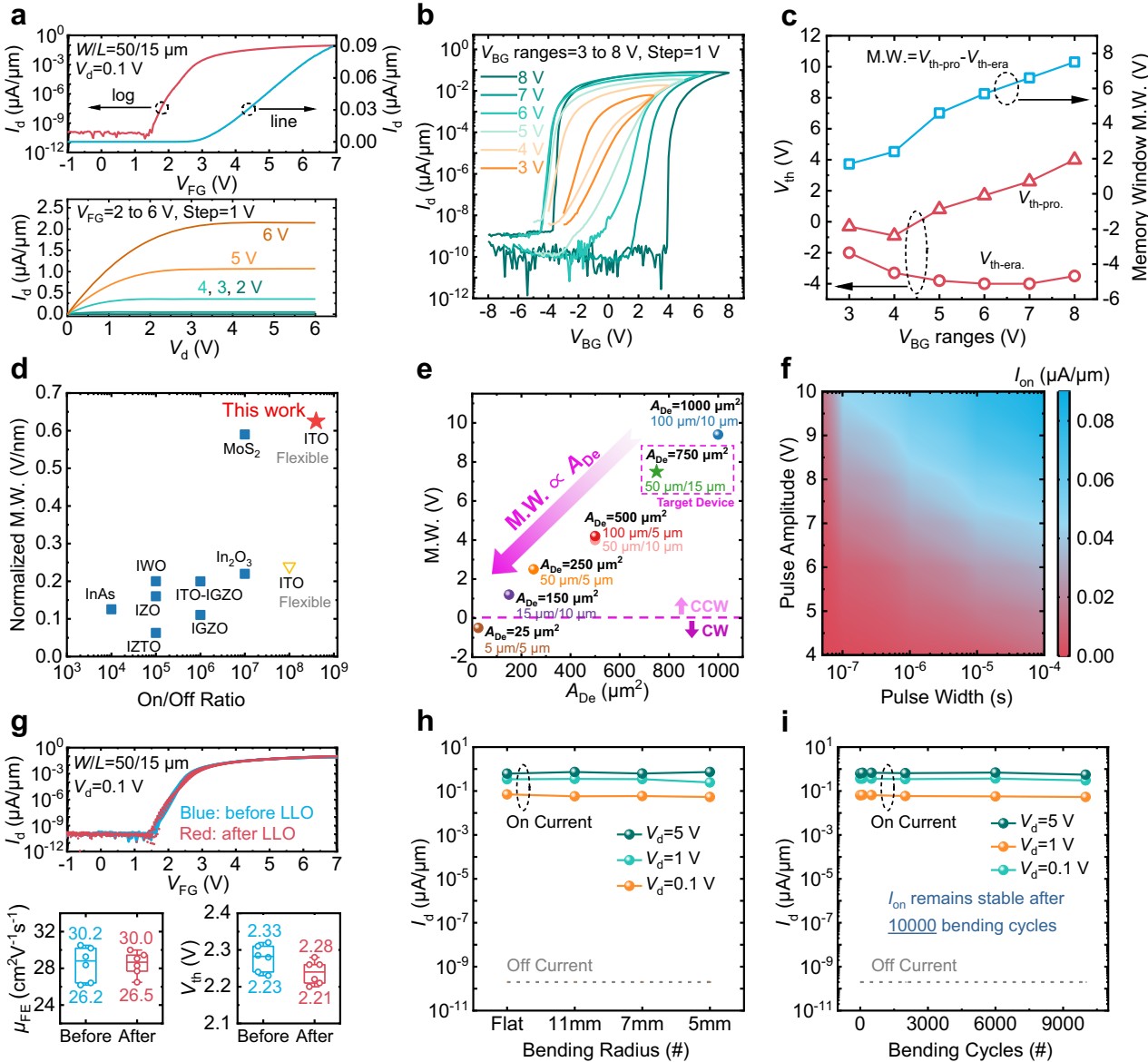

**Fig. 3 | Electrical characterization of flexible ITO FeMFET. a** Transfer curves at $V_d = 0.1\,V$ and output curves at $V_{FG}$ ranging from 2 V to 5 V of the ITO FeMFET. Hysteresis transfer curves (**b**) and extracted memory window (M.W.) (**c**) of the ITO FeMFET at various $V_{BG}$ ranges. **d** Benchmarking of normalized M.W. versus on/off ratio performance of the ITO FeMFET developed in this work (red star) and reported works recently (yellow triangle and blue square). A record normalized M.W. of 0.63 V/nm V (M.W. of 7.5 V) is obtained at $V_{BG}$ sweeps between ± 8 V. **e** M.W.

of ITO FeMFETs with a FeCAP area fixed at 9 μm² and varying $A_{De}$. **f** $I_{on}$ (at $V_d = 0.1\,V$) versus programming pulse amplitudes and pulse widths of the ITO FeMFET. **g** Typical transfer curves at $V_d = 0.1\,V$ of 6 ITO FeMFETs before and after LLO and boxplots of corresponding extracted key parameters. $I_{on}$ and $I_{off}$ at various $r_{bending}$ (**h**) and after different bending cycles (**i**) are measured. $I_{on}$ at various $V_d$ remains stable after 10000 bending cycles.

the driving current is achieved by varying these parameters, highlighting the potential for adaptive current supply in display systems. A transient switching measurement with pulse widths ranging from 10 ns to 500 μs (Supplementary Fig. 13) is performed. The pronounced enhancement in $I_{on}/I_{off}$ as the pulse width extends from 10 ns to 5 μs is primarily driven by FE-switching. The device also demonstrates robust endurance, maintaining stable operation over $5 \times 10^7$ cycles when subjected to ±8 V stress pulses at 200 kHz. Measurement pulses of ±10 V with 100 μs duration were applied to record the saturated $I_{on}$ values (Supplementary Fig. 14).

Figure 3g displays representative transfer curves and statistical distributions of key parameters for 6 ITO FeMFETs before and after the laser LLO process. The curves exhibit negligible shift or degradation following LLO. The boxplots of $\mu_{FE}$ and $V_{th}$ confirm consistent

distributions, indicating minimal process-induced variation. Figure 3h, i summarize $I_{on}$ and $I_{off}$ measured at $V_d = 0.1\,V$, 1 V, and 5 V under different bending radius and after repeated bending cycles. $I_{on}$ remains stable even at a bending radius of 5 mm and shows no degradation after $10^5$ cycles, confirming the mechanical robustness of the ultrathin ITO channel. $I_{off}$ remains at the measurement limit ($2 \times 10^{-10}$ μA/μm) across all conditions, a result of the wide memory window (7.5 V) inherent to the ITO FeMFET design.

## Demo of μ-LED display driven by 1T-1FeMFET

Figure 4a presents a schematic and micrograph of the 1T-1FeMFET pixel circuit integrated with a bonded μ-LED. The circuit consists of a conventional ITO TFT, acting as selecting transistor (Selecting-T, $W/L = 50$ μm/5 μm), and an ITO FeMFET, acting as the driving transistor

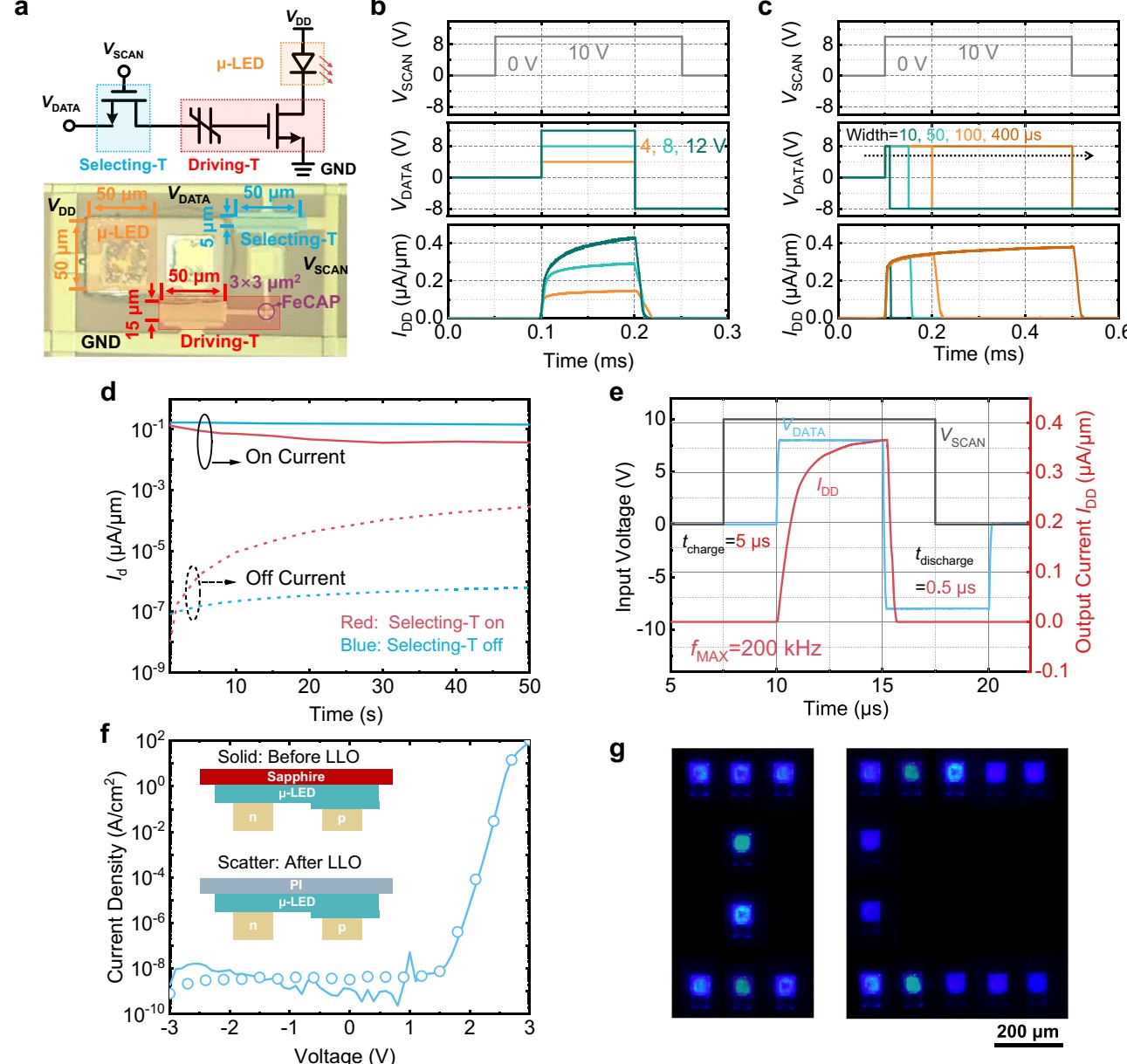

**Fig. 4 | Performance of 1T-1FeMFET pixel circuit and μ-LED display. a** The schematic and micrograph of 1T-1FeMFET pixel circuit with a bonded μ-LED. FeCAP features an ultra-small size of $3 \times 3\ \mu m^2$. **b** Driving current modulation of the pixel circuit under PAM scheme at $V_{data}$ from 4 V to 12 V. $I_{DD}$ is sensitive to the $V_{data}$ signal. **c** Driving current modulation under PWM scheme at pulse width of $V_{data}$ from 10 μs to 400 μs. **d** Retention characteristics the ITO FeMFET in the pixel circuit. The programmed or erased state is maintained for over 50 s due to the non-volatility of the HZO layer. **e** High-frequency characteristics of the ITO FeMFET in the pixel circuit. Both the great current capacity of the ITO selecting transistor and the low capacitance of the small size FeCAP contribute to a high pixel refresh rate of 200 kHz. **f** The *J-V* curves of the μ-LED before and after LLO. **g** The patterned images of letters "I" and "C" with the flexible AM μ-LED display.

(Driving-T, $W/L = 50\ \mu m/15\ \mu m$, FeCAP size = $3 \times 3\ \mu m^2$). The μ-LED dimensions are $50 \times 50\ \mu m^2$. The non-volatile character of the FeMFET effectively suppresses charge loss at the storage node, supporting the use of a significantly smaller capacitance than in conventional 2T-1C designs. Consequently, the output current of the Driving-T can be controlled via programming or erasing pulses of microsecond duration applied to the $V_{DATA}$ input. Supplementary Fig. 15 shows voltage sequence and corresponding static characteristics of the 1T-1FeMFET pixel circuit. The Selecting-T, which provides a charging path to the Driving-T, is controlled by $V_{SCAN}$ to modulate $I_{DD}$ via limiting the amplitude of $V_{DATA}$. The non-volatile memory function of the FeMFET stems from the aligned polarization in the FeCAP, which restraints charges on the metal electrodes on both sides of the ferroelectric layer, leading to a minimal leakage from floating gate, strengthening the charges retention ability (Fig. 1b). Consequently, the potential at the source of the Selecting-T, determined by both $V_{DATA}$ and $V_{SCAN}$, governs polarization switching in the HZO layer and sets the on-state current of the Driving-T ($I_{DD}$), as shown in Supplementary Fig. 15b.

Figure 4b demonstrates the pulse amplitude modulation (PAM) scheme for precise grayscale control. The $I_{DD}$ exhibits a strong dependence on the amplitude of $V_{DATA}$ (from 4 to 12 V) when $V_{SCAN}$ is held at 10 V, enabling effective current modulation of the Driving-T. To mitigate color shift in μ-LEDs, pulse width modulation (PWM) is also implemented. As shown in Fig. 4c, the $I_{DD}$ remains stable across

**Table 1 | Benchmarking of PPI and $P_{dynamic}$ of the proposed 1T-1FeMFET pixel circuit versus reported pixel circuits**

| | [49] | [50] | [51] | [52] | [53] | [54] | [55] | This work | |
|---|---|---|---|---|---|---|---|---|---|
| | | | | | | | | / | DTCO |
| Technology | LTPS | LTPS | LTPO | LTPO | IGZO | IGZO | ITO | ITO | ITO |
| Cell Structure | 6T2C | 8T1C | 9T3C | 6T2C | 2T1C | 2T1C | 2T1C | 1T-1FeMFET | 1T-1FeMFET |
| Capacitor Area (µm$^2$) | 100 | 80 | 90 | 100 | 960 | 40000 | 3800 | 9 | 0.09 |
| PPI (@4 inches, 16:9) | 481 | 479 | 479 | 484 | 404 | 123 | 303 | 428 | 4280 |
| $P_{dynamic}$ (of a pixel @60 Hz, nW) | 1.5 | 0.85 | 88 | 4.4 | 5.1 | 17 | 5.03 | 0.68 | 0.0068 |

Performance of the 1T-1FeMFET pixel circuit after DTCO by 10× area reduction has been predicted.

pulse widths of $V_{DATA}$ ranging from 10 to 400 µs, confirming reliable grayscale control through duty cycle variation. Figure 4d demonstrates the retention characteristics of the pixel circuit with the Selecting-T both on and off. When the Selecting-T is on and $V_{DATA}$ is held at 0 V, the programmed or erased state is maintained for over 50 s due to the non-volatility of the HZO layer. This confirms that the $V_{th}$ and leakage current of the Selecting-T requires no special design consideration, and the circuit even support long-term data storage without power. With the Selecting-T off after full polarization switching, $I_{on}$ also shows excellent stability over the same duration. A larger increase in $I_{off}$ is observed, driven by the ground potential at $V_{DATA}$ accelerating charge de-trapping, coupled with the activation of a discharge path.

Figure 4e characterizes the high-frequency response of the pixel circuit under $V_{DATA}$ = 8 V and $V_{SCAN}$ = 10 V, showing a pixel refresh rate of 200 kHz−sufficient to support a 90 Hz refresh rate at 4 K resolution. It is attributed to the great current capacity of the ITO selecting transistor and the low capacitance of the small size FeCAP of 9 µm$^2$. The $J$–$V$ characteristics and electroluminescence spectra of the µ-LEDs before and after LLO exhibit negligible variation, as illustrated in Fig. 4f and Supplementary Fig. 16. The on-wafer external quantum efficiency (EQE) of the µ-LED is presented in Supplementary Fig. 17. In Fig. 4g, predefined $V_{DATA}$ signals successfully drive a patterned display of the letters "I" and "C", demonstrating independent addressability of individual µ-LEDs. Optical image of the complete µ-LED display module with peripheral signal lines and micrograph of the active µ-LED display array are shown in Supplementary Fig. 5b, c.

A benchmarking analysis in Table 1 compares the pixels per inch (PPI) and dynamic power consumption ($P_{dynamic}$ = $C \cdot V^2 \cdot f_{refresh}$) of this work against previously reported pixel circuits[62–68]. Here, PPI is derived from the effective area of a single pixel, taken as the combined footprint of the driving transistor, selecting transistor, storage capacitor, and the µ-LED, under a defined 4-inch, 16:9, 60 Hz display configuration. This layout-independent approach enables a fair comparison across different pixel circuits. Owing to the minimal FeCAP area, the proposed design achieves the lowest $P_{dynamic}$ of 0.68 nW as a result of reduced charge storage. Based on the intrinsic non-volatility of the FeMFET, the proposed pixel architecture possesses exceptional scaling potential. While the current device dimensions are constrained by our experimental processing capabilities, the scaling potential of the architecture is evaluated via design-technology co-optimization (DTCO) based on mainstream fabrication linewidths used by industry leaders such as BOE Technology Co. Ltd. Adopting standard design rules−including proportional scaling of the FeMFET channel length and FeCAP area−to match advanced LTPS (for OLED) or oxide (for high-end LCD) technology nodes (channel lengths ~ 0.5–1 µm), a 10× reduction in pixel pitch from the current design is readily feasible. This scaling yields a projected pixel density exceeding 4280 PPI and a dropping $P_{dynamic}$ of 6.8 pW (Table 1). These projections underscore the strong potential of the 1T-1FeMFET architecture for next-generation high-resolution, ultra-low-power flexible displays.

## Discussion

In summary, we demonstrate fully flexible, 1T-1FeMFET driven AM µ-LED display with 428-PPI and ultra-low power consumption of 0.6 nW. The entire system was integrated on PI substrates at processing temperatures below 400 °C. Key to this achievement is the co-design of ferroelectric capacitors and transistors: the 9-µm$^2$ FeCAPs exhibit a high remnant polarization of 47 µC/cm$^2$, while the ITO-based FeMFETs show a record normalized memory window of 0.63 V/nm and an on/off ratio of $4 \times 10^8$, enables substantial reductions in both pixel size and power consumption. The pixel circuit enables precise grayscale control through pulse modulation (PAM/PWM) and stable driving under bending radius as small as 4 mm. Successful integration with GaN-based µ-LEDs confirms the architectural viability of our approach for future high-resolution, energy-efficient, and truly flexible display technologies for next-generation wearable and portable electronics.

## Methods
### Fabrications of ITO FeMFET

Supplementary Fig. 18a shows the key fabrication process of the ITO FeMFETs. The ITO FeMFETs were fabricated on the PI film which is adhered to a glass carrier (Wuxi Alflex Optoelectronic Technology Co., Ltd). The PI film exhibits excellent processing compatibility, characterized by key properties including low surface roughness, a thickness of 38 µm, thermal shrinkage <0.1% (after 2 h at 400 °C), a coefficient of thermal expansion (CTE) of 3 ppm/°C, and a maximum tolerable temperature of 450 °C (specifications provided by the supplier). Given this 450 °C limit, the process temperature is set to 400 °C to ensure reliable substrate performance. The substrate was pre-heated at 400 °C for 2 h to enhance thermal stability after cleaning. A SiO$_2$ buffer layer was deposited via PECVD, followed by sputtering a stack of 50-nm Ti and 50-nm Mo as bottom gate (BG). Then, a 12-nm Hf$_{0.5}$Zr$_{0.5}$O$_2$ (HZO) layer was deposited by ALD at 220 °C, utilizing HF, TDMAZ (C$_8$H$_{24}$N$_4$Zr), and H$_2$O as precursor. Then a layer of 100-nm Mo serving as floating gate (FG) was sputtered. After that, a post-metallization annealing (PMA) at 400 °C for 2 h is conducted to crystallize the HZO layer, a temperature chosen to balance the maximum tolerable temperature of the flexible polyimide substrate with the need for optimal ferroelectric performance (Supplementary Fig. 19). Oxygen adsorption by Mo enhances the ferroelectric properties of HZO sandwiched between two Mo layers during post-metallization annealing (PMA). Then, 20-nm Al$_2$O$_3$ serving as gate dielectric (GI) was deposited via ALD at 250 °C. A 6-nm ITO (In$_2$O$_3$:SnO$_2$ = 95:5 wt%) channel layer was sputtered with the Ar/O$_2$ ratio, power, and working pressure of 45/5 sccm, 60 W, and 3 mTorr, respectively. 60-minutes annealing was conducted at 200 °C in the air condition. Finally, 100-nm Mo was sputtered and patterned as the source/drain (S/D) contacts. Finally, SU-8 was spin-coated for passivation layer and contact holes were achieved through photolithography.

### Transfer of µ-LEDs
Supplementary Fig. 18b shows the transfer of µ-LEDs. The µ-LEDs were fabricated on a sapphire substrate through sequential steps of etching

mesa, Ti/Al/Ni/Au electrodes evaporation, and $SiO_2$ passivation. A UV tape was attached to the surface of the μ-LEDs and then they were laser-lift-off (LLO) from the sapphire substrate. After that, the μ-LEDs with UV tape were adhered to another PI tape. Then, a UV irradiation was conducted to reduce the adhesion of the UV tape and remove it.

## Preparation of flexible display
Supplementary Fig. 18c shows the preparation of flexible display. After preparing the ITO FeMFETs on a glass carrier and the μ-LEDs array on a PI film, the μ-LEDs array was bonded to the ITO FeMFETs driving backplane. Then, the PI film on the surface of μ-LEDs was removed mechanically. A 308-nm excimer laser was utilized to separate the PI film from the glass carrier. Finally, the 1T-1FeMFET flexible display was demonstrated.

## Device characterizations
The electrical properties of ITO FeMFETs and μ-LEDs were examined using Agilent B1500A semiconductor device parameter analyzer. The ferroelectric characteristics were tested using Keithley 4200A-SCS unit (Tektronix) and CRX-6.5 K cryogenic probe station (Lake Shore). The transient response of the 1T-1FeMFET pixel circuit were measured using Keithley 4200A-SCS unit and HSPY-1500-002 adjustable DC regulated switching power supply. The electroluminescent spectra of the μ-LEDs was measured using NOVA2S-EX fiber optic spectrometer (ideaoptics). The TEM images were obtained using FEI Titan 80−300 scanning transmission electron microscope (STEM).

## Data availability
The Source Data underlying the figures and tables of this study are available with the paper. All raw data generated during the current study are available from the corresponding authors upon request. Source data are provided with this paper.

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

## Acknowledgements

The authors acknowledge support from the National Natural Science Foundation of China (62274033 to W.W., 62404110 to G.Y., 62504040 to R.S., 62274083 to Z.Z., T2221003 to Z.Z.), Natural Science Foundation of Jiangsu Province (BK20232042 to Z.Z.), Fundamental Research Funds for the Central Universities (2242025K30014 to R.S.), and Natural Science Research Start-up Foundation of Recruiting Talents of Nanjing University of Posts and Telecommunications (NY223159 to G.Y.).

## Author contributions

W.W., R.S., W.S., B.L., and Z.Z. conceived and supervised the project. T.H., G.Y., Y.S., F.Z., M.L., G.L., H.C., Zuoxu Y., Z.F., S.L., Zhihao Y., X.W., and H.S. contributed to sample preparation, characterization, device fabrication and data analysis. T.H., M.L., G.L., and H.C. performed the measurements. T.H., G.Y., and Y.S. co-wrote the manuscript with input from other authors. All authors contributed to discussions.

## Competing interests

The authors declare no competing interests.
