## [Transparent Peer Review file · Nature Communications]

Flexible Active-Matrix Micro-LED Display with 1T-1FeMFET Architecture Featuring Scaling-Limit-Free Design

Corresponding Author: Professor Wangran Wu

Version 0:

Reviewer comments:

Reviewer #1

(Remarks to the Author)

The paper demonstrates the integration of HZO based FeMFET cells as pixel driver for μ LED applications. It shows promising and benchmarking results about this kind of integration also utilizing the FE capacitor to fix charges on the floating node. However, I suggest a revision for pointing out individual points more clearly.

Remarks for individual lines:

L44: Rely instead of reply?

L61-62: In many publications (that are not cited here), it has already been demonstrated that crystallization of HZO below 400C is not a challenge anymore.

L155: How can a voltage be applied to the FG? There is not direct connection in the circuit diagram shown. Please comment on this.

178: Record MW: I suggest to discuss deeper and clearer that this is related to a mixture of FE-switching and FG-Charging. With charging as the main mechanism as you state later. It would be good also to calculate/estimate & validate based on the formular at which voltage the HZO switches.

190: Figure 3f shows that switching happens with very long pulses suggesting that main mechanism is charge-trapping into floating node rather than FE-switching

S Picture 2: is the E-field correct... we know about switching at an order of magnitude higher fields

General remark: A micrograph of the entire structure including scale would be beneficial.

Reviewer #2

(Remarks to the Author)

This manuscript reports 1T-1FeMFET pixel circuit by incorporating ITO channel and HZO gate stack, suitable for flexible substrates with low processing temperature, high field-effect mobility and low off-currents. The pixel circuit reported by Huang et al. presents competitive statistics with remnant polarization, record memory window, on/off ratio, flexible durability and application to μ LED. We consider this manuscript suitable for publication in Nature Communications, after provided with reply to following comments regarding detailed reasoning for the fabrication and results.

1. Details on processing temperature: the authors stated that while HZO-based FeCAPs and FeFETs provide promising platform, the high crystallization temperature limits the practical use on flexible substrates. Thus, in this work ITO channels with lower processing temperatures < 200 °C are used. However, pre-heating of the substrates and processing of HZO layer in the FeMFET presented in this work still requires 400 °C of processing. Is 400 °C the optimal temperature, and what happens if lower or higher temperature is used to Pr of flexible FeCAP. If the device can endure the temperature and higher temperatures provide higher Pr, that should also be mentioned in the manuscript.

2. Details on channel dimension:

- The authors used channel dimensions of 50/15 μm for ITO FeMFET. While the superiority of 1T-1FeMFET drives from allowing miniaturization of the capacitors, the reduction in channel dimension should also be required. Thus, trend of channel reduction in ITO FeMFET should be tested, as well as mentioned in Table S2 for comparison.

- Also, the details on projected scalability and the projected pixel density of μLED should be provided, and if they can be confirmed experimentally.

3. Details on record normalized memory window: as presented by Table S2, ITO FeMFET presented by the authors demonstrate record M.W. compared to previous studies. While same HZO is used for ferroelectricity, different channel materials are used ranging from oxides to 2D materials. More discussion should be mentioned in the manuscript as to why ITO gives higher M.W. in comparison to other channel materials, what defines the mechanism to the resultant M.W. and why that differs between the chosen materials.

4. Details on the low I_{off} , line 204 mentioned that I_{off} remains at the measurement limit across all condition, which is at $2 \times 10^{-10} \mu\text{A}/\mu\text{m}$. If this value is extracted due to machine limitation (compliance), is it truly correct value? Please elaborate on this value, and if more accurate data can be extracted by change of device dimensions or alternative measurement.

5. Details on Figures

- Figure 2c. P-V hysteresis loop at various voltage ranges, the voltages are not labeled

- Figure S4. Provide scale

- Figure 4. Provide emission efficiency

6. The authors miss the important work in oxide transistors field reported recently. I recommend that the author refer to the following works: such as Nature 629 (8013), 798-802, Science Advances 11 (43), eadz6914

Reviewer #3

(Remarks to the Author)

This work presents a new configuration integrating FeMFET and ITO channel into a 1T-1FeMFET pixel circuit for flexible active matrix $\mu\text{-LED}$ display application. The scheme leverages on the merging of a driving transistor and an embedded memory element, allowing to scale down the pixel size. The FeMFET demonstrated record normalized window as well as impressive stability under mechanical stress when bent down to a radius as small as 4 mm. The work further presented extensive characterization of the device and its employment for flexible displays. The following comments would likely improve the quality of the paper:

1. Lines 60-65. It is unclear why the high-temperature requirement for HZO deposition/functionalization needs to be mentioned as a disadvantage. It is understood that it is not replaced by ITO, which is processable at lower temperatures.

2. What is the thickness of the polyimide film on glass? Polyimide films typically have a maximum workable temperature around 250°C – 275°C . Could you please comment on the thermal expansion experienced by the flexible substrate used after performing the steps requiring a high temperature of 400°C ?

3. What are the critical interfaces/layers during bending? For the mechanical stability test performed by bending, please report the estimated induced strain. How does the crystallinity of HZO affect or support the mechanical performance of the circuit?

4. Please add a comment on the presence of hump in the transfer characteristics Fig. 3a and 3g.

5. In Fig. 4d, there is more than 4 orders of magnitude increase in the off-current of the transistor when the selecting transistor is on. What does this increase imply?

Version 1:

Reviewer comments:

Reviewer #1

(Remarks to the Author)

I reviewed the the feedback and updates of the authors. Many thanks for carefully considering and discussing my comments which completes the article very nicely from my point of view. I have no further comments and suggest publication.

Reviewer #2

(Remarks to the Author)

The paper is properly revised, I recommend this paper will be accepted as it is.

Reviewer #3

(Remarks to the Author)

The authors have satisfactorily and elaborately responded to all the comments and provided the necessary modifications to the manuscript. Therefore, I recommend the manuscript for publication as it is.

Reviewer 1:

The paper demonstrates the integration of HZO based FeMFET cells as pixel driver for μ LED applications.

It shows promising and benchmarking results about this kind of integration also utilizing the FE capacitor to fix charges on the floating node. However, I suggest a revision for pointing out individual points more clearly.

Comment 1:

L44: Rely instead of reply?

Our response:

Thanks for the comment. We have replaced the word “reply” with “rely” in the revised version.

Our modification:

➤ **Line 44 on page 2 in the manuscript:** We have replaced the word “reply” with “rely”.

Comment 2:

L61-62: In many publications (that are not cited here), it has already been demonstrated that crystallization of HZO below 400C is not a challenge anymore.

Our response:

Thank you for the comment. The crystallization of HZO at temperatures below 400 °C has been widely reported in the literature [1–21] and is summarized in Table R1. Figure R1a benchmarks the relationship between remanent polarization ($2P_r$) and annealing temperature for the proposed FeCAPs in this work (red star) against recently reported results. As shown, the ferroelectric properties of most previously reported devices remain limited at such low processing temperatures, as reflected by their relatively modest $2P_r$ values. These results indicate that achieving robust ferroelectric characteristic with large remanent polarization ($2P_r$) remains challenging under low-temperature processing constraints. In this context, the $2P_r$ achieved in this work approaches the highest values reported within a compatible temperature window, highlighting the effectiveness of the adopted fabrication flow and post-metallization annealing (PMA) process.

Then we explain the motivation for selecting 400°C as the PMA temperature. The selection of a 400°C annealing temperature was driven by a balance between the maximum tolerable temperature of the flexible substrate and the crystallization requirement of HZO. Specifically, the polyimide (PI) substrate used in this work has a specified maximum tolerable temperature of 450°C. To ensure the mechanical reliability of the substrate, including its flexibility, dimensional stability, and low residual stress, throughout the full integration flow, a safe thermal margin should be preserved. Furthermore, the polarization of ferroelectric capacitors (FeCAPs) generally exhibits a positive correlation with annealing temperature up to ~ 600°C [15-25]. It is also confirmed in our prior experimental data, as shown in Fig. R1b. To enhance both the remnant polarization $2P_r$ of the FeCAPs and the resulting memory window of the ITO FeMFET, it is crucial to apply the highest feasible temperature without compromising the stability of the PI substrate. Thus, the HZO crystallization anneal was conducted at 400°C.

Figure R1: (a) Benchmarking of $2P_r$ versus annealing treatment temperature of the proposed FeCAPs in this work (red star) and recently reported works. (b) P-V hysteresis loops expand with increasing PMA temperature.

Table R1: Benchmarks of process temperature and $2P_r$ of FeCAPs reported in this work versus recently reported FeCAPs.

Structure	FE	Thickness of FE (nm)	Material of flexible substrate	Annealing process	Process temperature (°C) and time	$2P_r$ ($\mu\text{C}/\text{cm}^2$)	Year	Ref.
Mo/HZO/Mo	HZO	30	Mica	FMA (Focused-microwave Annealing)	250, /	40	2022	2
TiN/HZO/TiN	HZO	10	Si	RTA (Rapid Thermal Annealing)	400, 60s	50	2022	4
TiN/HZO/TiN	HZO	10	Si	FA (Furnace Annealing)	300, 48h	30	2025	5
TiN/HZO/TiN	HZO	10	Si	RTA	350, 30s	26.9	2024	6
Mo/HZO/Mo	HZO	10	Si	EA (Exhalative Annealing)	200, 0.5h	34.4	2024	7
InAs/HZO/TiN	HZO	12	InAs	RTA	370, /	42	2020	10
TiN/HZO/TiN	HZO	10	Si	FA	400, 1h	36	2022	11
Pt/HZO/TiN	HZO	7	Si	ALA (Atomic Layer Annealing)	300, /	42.6	2022	12
TiN/HZO/TiN	HZO	10	Polyimide	RTA	400, 60s	21	2021	13
TiO _x /HZO/Mo	HZO	12	Si	RTA	400, 30s	36	2023	14
TiN/Al ₂ O ₃ /HZO/TiN	HZO	6	Si	RTA	400, 30s	31	2024	15
TiN/HZO/TiN	HZO	8.5	Si	HPA (High-Pressure Annealing)	300, 0.5h	26	2021	16
TiN/HZO/TiN	HZO	10	Si	RTA	400, 60s	40	2020	17
TiN/HZO/TiN	HZO	7	Si	RTA	400, 60s	40.5	2018	18
W/HZO/W	HZO	6	Si	RTA	400, 60s	43.4	2025	19
TiN/HZO/Ge	HZO	10	Ge	RTA	300, 30s	38	2024	20
TiN/HZO/TiN	HZO	10	Si	FLA (Flash Lamp Annealing)	375, /	42	2018	21
TiN/ZrO ₂ /HZO/SiO ₂	HZO	10	Si	RTA	300, 60s	15.2	2022	22
TiN/HZO/TiN	HZO	7	Si	RTA	300, 60s	12.7	2025	23
Pt/HZO/TiN	HZO	6	Si	RTA	400, 60s	51.2	2021	24
TiN/HZO/TiN	HZO	10.2	Si	Ultraviolet-LED	400, /	50	2024	25
Mo/HZO/Mo	HZO	12	Polyimide	PMA (Post Metallization Annealing)	400, 2h	47	2025	This work

Our modification:

- **Line 29-30 on page 2 in the manuscript:** We have replaced the sentence “The low-temperature ferroelectric capacitors (FeCAPs) in FeMFETs exhibit a remnant polarization ($2P_r$) of $47 \mu\text{C}/\text{cm}^2$.” with “**The 400°C-activated ferroelectric capacitors (FeCAPs) in FeMFETs exhibit a remnant polarization ($2P_r$) of $47 \mu\text{C}/\text{cm}^2$.**”.
- **Line 61-64 on page 3 in the manuscript:** We have replaced the sentence “However, the high crystallization temperature required for HZO limits the fabrication and performance of FeMFET on flexible substrates²⁴⁻²⁸.” with “**However, reports on excellent ferroelectric properties in HZO and the subsequent high-performance FeMFETs on flexible substrates remains scarce, compounded by a lack of systematic studies on their electrical and mechanical properties. These limitations pose a key constraint on the development of FeMFETs for flexible electronics²⁴⁻²⁸.**”.
- **Line 79 on page 3 in the manuscript:** We have replaced the sentence “on a PI substrate with

thermal budget below 400°C.” with “on a PI substrate with processing temperature below 400°C.”.

- **Line 124-127 on pages 4 and 5 in the manuscript:** We have replaced the sentence “As shown in Fig. 2c, the ferroelectric polarization undergoes complete switching at voltages above 3 V. Partial polarization switching can be achieved by reducing the sweep amplitude, as further illustrated in Supplementary Fig. S3.” with “As shown in Fig. 2c, the ferroelectric polarization undergoes complete switching at voltages above 3 V, although this yields insufficient $2P_r$ for FeCAPs after 300°C PMA. Reducing the sweep amplitude enables partial polarization switching (Supplementary Fig. S3).”.
- **Line 328-331 on page 10 in the manuscript:** We have replaced the sentence “After that, a post metallization annealing (PMA) at 400°C for 2 hours is conducted to crystallize HZO.” with “After that, a post-metallization annealing (PMA) at 400°C for 2 hours is conducted to crystallize the HZO layer, a temperature chosen to balance the maximum tolerable temperature of the flexible polyimide substrate with the need for optimal ferroelectric performance (Supplementary Fig. S19).”.
- **Figure 2 in the manuscript:** We have updated Fig. 2c to more clearly show the positive correlation of ferroelectricity with annealing temperature.
- **Figure 19 in the supplementary:** We have added Fig. R1b as Supplementary Figure 19 to clarify the enhanced ferroelectric performance of FeCAPs achieved with 400°C PMA compared to 300°C PMA.

Comment 3:

L155: How can a voltage be applied to the FG? There is not direct connection in the circuit diagram shown. Please comment on this.

Our response:

We thank the reviewer for pointing out the lack of a direct electrical connection to the floating gate (FG) in the original circuit diagram. To address this concern, we have revised Fig. 1 to illustrate how the FG and bottom gate (BG) are accessed and biased in the ITO FeMFET within the 1T-1FeMFET pixel circuit (updated as Fig. R2a). Additionally, we now provide a more detailed explanation of the two operational modes of the ITO FeMFET, ferroelectric mode (Fe-mode) and dielectric mode (De-mode), as illustrated in Figs. R2b and R2c.

In Fe-mode, as shown in Fig. R2b, a DC sweep or voltage pulse is applied to the BG (V_{BG}) while the FG is left floating, the source is grounded, and a constant voltage is supplied to the drain (V_D). Under these conditions, polarization in the HZO layer directly modulates the conductivity and carrier density of the ITO channel, thereby enabling memory operation.

In De-mode, as shown in Fig. R2c, a DC sweep or voltage pulse is applied to the FG (V_{FG}) with the BG floating, the source grounded, and a constant V_D applied. Here, the influence of ferroelectric polarization is screened, allowing the device to function as a conventional ITO TFT.

Figure R2: (a) Revised Fig. 1 in the manuscript. Detailed operating explanation of (b) Fe-mode and (c) De-mode for the proposed ITO FeMFET.

Our modification:

- **Line 161 on page 6 in the manuscript:** We have replaced the sentence “supporting operation in two distinct modes.” with “supporting operation in two distinct modes (Supplementary Fig. S6).”.
- **Figure 6 in the supplementary:** We have added Figs. R2b and c as Supplementary Figure 6 to better illustrate the Fe-mode and De-mode operations.
- **Figure 1 in the manuscript:** We have replaced Fig. 1 with Fig. R2a to more clearly show the connection of the ITO FeMFET within the circuit diagram.

Comment 4:

178: Record MW: I suggest to discuss deeper and clearer that this is related to a mixture of FE-switching and FG-Charging. With charging as the main mechanism as you state later. It would be

good also to calculate/estimate & validate based on the formular at which voltage the HZO switches.

Our response:

We are grateful to the reviewer for this constructive feedback. We agree that the record M.W. in our manuscript is indeed the results of the synergistic effect of ferroelectric (FE) switching and charge trapping. Thus, we will provide an explicit discussion of the memory mechanism and supplement the relevant sections in the manuscript accordingly.

1. Contribution from FE-switching

According to the traditional FE-based model, the memory window of FeFETs arises primarily from polarization reversal. Therefore, the M.W. is obtained through the following calculation:

We begin with the law of charge conservation. It is worth noting that the total charges within the ferroelectric HZO layer (Q_{FE}) consist of polarization (P_{HZO}) and non-polarization ($P_{HZO,non}$). Thus, due to the series of capacitance of the HZO layer and the capacitance of the Al_2O_3 layer, we can obtain

$$Q_{\varphi} = Q_{FE} = P_{HZO} + P_{HZO,non} = P_{HZO} + C_{HZO} \cdot V_{HZO} \quad (1)$$

where C_{HZO} and V_{HZO} are the capacitance and voltage drops of the HZO layer, and Q_{φ} is the charges in the ITO layer, which is associated with the surface potential (φ) of the ITO channel. For Fe-mode operation of the ITO FeMFETs, we apply a DC sweeping voltage to the BG (V_{BG}), leave the FG floating, keep the source contact grounded ($V_s = 0$ V), and apply a constant voltage to the drain contact (V_d). Therefore, the applied V_{BG} is distributed across the HZO ferroelectric layer, the Al_2O_3 dielectric layer, and the ITO channel, and can thus be expressed as

$$V_{BG} = V_{HZO} + V_{DE} + \varphi \quad (2)$$

where V_{DE} is the voltage drop in the Al_2O_3 layer. Thus, the threshold voltage (V_{th}) of the programmed ($V_{th-pro.}$) and erased states ($V_{th-era.}$) can be expressed as

$$V_{th-pro.} = V_{HZO-pro.} + V_{DE-pro.} + \varphi_{pro.} \quad (3)$$

$$V_{th-era.} = V_{HZO-era.} + V_{DE-era.} + \varphi_{era.} \quad (4)$$

where V_{HZO} , V_{DE} , and φ , with the subscripts “-pro.” and “-era.”, denote the voltage drops across the HZO layer and Al_2O_3 layer and the surface potential of the ITO channel in the programmed and erased states, respectively. Then we can obtain

$$M.W. = V_{th-pro.} - V_{th-era.} = (V_{HZO-pro.} - V_{HZO-era.}) + (V_{DE-pro.} - V_{DE-era.} + \varphi_{pro.} - \varphi_{era.}) \quad (5)$$

Since at threshold, the voltage drops across the Al_2O_3 layer and the surface potential of the ITO channel are identical for both the programmed and erased states. Considering formula (1), the M.W. can be expressed as follows

$$M.W. = V_{HZO-pro.} - V_{HZO-era.} = \frac{Q_{\varphi-pro.} - P_{HZO-pro.}}{C_{HZO}} - \frac{Q_{\varphi-era.} - P_{HZO-era.}}{C_{HZO}} \quad (6)$$

where Q_{φ} and P_{HZO} , with the subscripts “-pro.” and “-era.”, denote the charges in the ITO layer and the polarization within the HZO layer in the programmed and erased states, respectively. Furthermore, owing to the identical charge concentration in the ITO channel at the threshold

condition, the corresponding charge terms $Q_{\phi\text{-pro.}}$ and $Q_{\phi\text{-era.}}$ are equal. So, the expression for the M.W. reduces to

$$M.W. = V_{HZO\text{-pro.}} - V_{HZO\text{-era.}} = \frac{P_{HZO\text{-era.}} - P_{HZO\text{-pro.}}}{C_{HZO}} \quad (7)$$

Typically, M.W. is bounded by $2V_C$ (twice the coercive voltage of the FE layer). However, our measured M.W. exceeds this limit, indicating an additional contributing mechanism.

2. Role of Charge Trapping

Our ITO FeMFET possesses a high area ratio ($AR \approx 83$) between the FeCAP ($9 \mu\text{m}^2$) and the ITO TFT ($750 \mu\text{m}^2$). This results in most of the applied V_{BG} dropping across the metal–ferroelectric–metal (MFM) capacitor, with only a small fraction across the ITO TFT. The voltage applied to the MFM capacitor and ITO TFT will trigger the tunneling current flowing inside these two parts, corresponding to the charge trapping in FG contributed by the ITO channel direction and BG direction, respectively [26-30], as illustrated in the energy-band diagrams (Figs. R3a and b). However, due to the considerable thickness of the Al_2O_3 layer (20 nm) with the relatively small voltage dropped across it, the effect of transferred charges between the ITO and FG is overwhelmed by that between the BG and FG. Consequently, the charge trapping is primarily governed by the charge transfer between BG and FG. At the onset of the backward sweep, the positive applied voltage $+V_{BG}$ attracts the electrons trapped in the FG to flow toward the BG direction, leading to a negative shift in the V_{th} . Conversely, at the beginning of the forward sweep, the negative applied voltage $-V_{BG}$ repels electrons from the BG region, resulting in a positive shift in the V_{th} . Both of these charge-trapping processes effectively widen the overall memory window [26-30].

3. Synergistic Effect

The record normalized memory window of 0.63 V/nm (7.5 V absolute) observed in our ITO FeMFET therefore stems from the synergistic effect between FE-switching and charge trapping. Owing to the high AR of 83, the majority of the V_{BG} drops across the HZO layer, thus enabling complete polarization reversal. This FE-switching yields a fundamental M.W. of approximately 3 V (approaching $2V_C$, as can be derived from Fig. 2c). Subsequently, charge trapping further expands the M.W. beyond the conventional $2V_C$ limit to 7.5 V. Therefore, while both mechanisms contribute, charge trapping is the more critical mechanism for achieving the recorded M.W..

To validate the aforementioned synergistic effect, we tested the hysteresis transfer curves of ITO FeMFETs with a fixed channel dimension (A_{De}) of $750 \mu\text{m}^2$ while varying FeCAP areas (A_{Fe}). As shown in Figs. R3c and d, the M.W. increases monotonically as A_{Fe} is scaled up from 9 to $400 \mu\text{m}^2$ (corresponding to an area ratio decrease from 83 to 1.9). This trend is attributed to the reduced voltage drop across the metal-ferroelectric-metal (MFM) capacitor at larger A_{Fe} . Consequently, the P-V hysteresis loop becomes unsaturated, yielding weaker polarization and less efficient charge transfer between the BG and FG. For the smallest A_{Fe} device, ferroelectric behavior is significantly diminished, owing to minimal polarization reversal and a negligible tunneling current.

Figure R3: The energy band diagrams of the charge trapping in the FG under (a) positive and (b) negative applied voltage. (c) Hysteresis transfer curves and (d) extracted M.W. of the ITO FeMFET as a function of the AR, with a fixed channel dimension (A_{De}) of $750 \mu\text{m}^2$.

Our modification:

- **Line 189-203 on pages 6 and 7 in the manuscript:** We have replaced the sentence “Record normalized M.W. is attributed to remarkable $2P_f$ and large potential across the HZO layer (V_{HZO}), which is expressed as

$$V_{HZO} = V_{BG} \times \frac{C_{DE}}{C_{DE} + C_{FE}} \quad (1)$$

where C_{FE} and C_{DE} are the capacitance of the HZO and Al_2O_3 layer, respectively²⁷. The memory properties of the ITO FeMFET can further enhance by modulating the ratio of C_{FE}/C_{DE} .” with “The record M.W. of 0.63 V/nm (7.5 V absolute) stems from the synergistic effect between ferroelectric (FE) switching in the HZO layer and charge trapping at the floating-gate (FG) node. With a channel dimension (A_{De}) of $750 \mu\text{m}^2$ and a FeCAP area (A_{Fe}) of $9 \mu\text{m}^2$, the resulting high area ratio ($AR = A_{De}/A_{Fe}$) of 83 causes the majority of V_{BG} to drop across the HZO layer, thus enabling complete polarization reversal. This FE-switching yields a fundamental M.W. of approximately 3 V, which approaches twice the coercive voltage ($2V_C$) of the HZO layer. Beyond this, the considerable thickness (20 nm) of the Al_2O_3 layer, along

with a relatively small voltage dropped across it, results in the charge transfer effect between the BG and FG overwhelming that between the ITO and FG (Supplementary Fig. S8). Subsequently, charge trapping further expands the M.W. beyond the conventional $2V_C$ limit to 7.5 V^{57-61} . Therefore, these two processes act synergistically to achieve the recorded M.W.. Besides, the hysteresis transfer curves of the ITO FeMFETs with a fixed A_{De} of $750 \mu\text{m}^2$ while varying A_{Fe} are investigated (Supplementary Fig. S9), which also serves to validate the aforementioned synergistic effect.”.

- **Figure 8 in the supplementary:** We have added Figs. R3a and b as Supplementary Figure 8 to depict the mechanism of charge trapping.
- **Figure 9 in the supplementary:** We have added Figs. R3c and d as Supplementary Figure 9 to validate the synergistic effect.
- **Refs. [57], [58], [59], [60], [61] on page 15 in the manuscript:** We have added reference [R26], [R27], [R28], [R29], [R30] as the reference [57], [58], [59], [60], [61].

Comment 5:

190: Figure 3f shows that switching happens with very long pulses suggesting that main mechanism is charge-trapping into floating node rather than FE-switching.

Our response:

We sincerely thank the reviewer for this valuable observation. A more detailed explanation of charge-trapping is provided below.

First, to clarify the dominant mechanism during pulse modulation testing, we performed a transient switching measurement on the ITO FeMFET (Fig. R4a). Using the SPGU module of a Keysight B1500A, a voltage sequence with pulse widths ranging from $500 \mu\text{s}$ to 10 ns (Fig. R4b) was applied to the BG while the FG was left floating, the source was grounded, and the drain was held at 0.1 V . As shown in Fig. R4c, I_d responds rapidly to both programming and erasing pulses and then remains stable for seconds. Moreover, the I_{on}/I_{off} ratio increases with longer pulse widths (Fig. R4d). Notably, a pronounced enhancement in I_{on}/I_{off} occurs when the pulse width extends from 10 ns to $5 \mu\text{s}$, a duration too short for significant charge trapping. This confirms that the improvement is primarily driven by FE-switching. Above $5 \mu\text{s}$, the increase in I_{on}/I_{off} continues but at a markedly slower rate, suggesting that the contribution from FE-switching approaches saturation, while the contribution from charge trapping at these longer timescales remains weak.

Second, the coupling between ferroelectric switching and charge trapping can be effectively decoupled by modulating the programming pulse width (W_{pro}), as established in previous studies [29-31]. The FE-switching dominates at microsecond-scale pulse widths, whereas charge trapping contributes substantially only when W_{pro} extends into the millisecond range. Therefore, within the width of programming pulse shorter than $100 \mu\text{s}$ (shown in Fig. 3f), FE-switching remains dominant rather than charge-trapping.

Finally, the measured series resistance of the probes and equipment is approximately 100Ω . In combination with the series capacitance of the HZO and Al_2O_3 layers, this yields an RC time constant on the order of $\sim 1 \text{ ns}$, indicating that electrical delay is not a limiting factor in our measurements. The polarization switching speed extracted for the device (Fig. 3f) is not slow.

Moreover, the switching performance we observe is consistent with, and in many cases comparable to or faster than, values reported in recent studies on HZO-based ferroelectric devices, as shown in Fig. R4e [31-36]. The data highlighted in pink region demonstrate that the devices still exhibit measurable changes in current or threshold voltage even under programmed/erased pulses with pulse widths as long as 10^{-4} to 10^{-3} s.

[32] Chen, Z. et al. 2024 IEEE International Electron Devices Meeting (IEDM).

[33] Duan, J. et al. 2024 IEEE IEDM.

[34] Huang, Z. R. et al. 2024 IEEE IEDM.

[35] Joh, H. et al. 2024 IEEE IEDM.

[36] Kim, G. et al. 2024 IEEE IEDM.

[37] Kuk, S.-H. et al. 2024 IEEE IEDM.

Figure R4: (a) Schematic of the transient switching measurement for the ITO FeMFET and (b) the measuring voltage sequence. Response of (c) I_d and (d) I_{on}/I_{off} for the ITO FeMFET to programming and erasing pulses with pulse widths ranging from 500 μ s to 10 ns. (e) Switching speed versus pulse amplitudes and pulse widths reported recently.

Our modification:

- **Line 219-222 on page 7 in the manuscript:** We have added the sentence “A transient switching measurement with pulse widths ranging from 10 ns to 500 μ s (Supplementary Fig. S13) is performed. The pronounced enhancement in I_{on}/I_{off} as the pulse width extends from 10 ns to 5 μ s is primarily driven by FE-switching.”.
- **Figure 13 in the supplementary:** We have added Figs. R4a, b, c, and d as Supplementary Figure 13 to illustrate that the device response under microsecond pulses is primarily governed by FE-switching.

Comment 6:

S Picture 2: is the E-field correct... we know about switching at an order of magnitude higher fields.

Our response:

The authors appreciate the reviewer’s insightful comment. The electric field values were erroneously reported in Supplementary Fig. S2 due to a miscalculation, which underestimated them by an order of magnitude. The values have been amended accordingly (Fig. R5).

Figure R5: Polarization switching depends on the electric field across the HZO layer (E_{HZO}).

Our modification:

- **Figure 3 in the supplementary:** We have replaced Supplementary Figure 3 with Fig. R5.

Comment 7:

General remark: A micrograph of the entire structure including scale would be beneficial.

Our response:

We express our gratitude to the reviewer for this comment. We have updated Fig. 4a with detailed size in the circuit image, as shown in Fig. R6a. The proposed 1T-1FeMFET pixel circuit consisted of a conventional ITO TFT with the channel width and length (W/L) of 50 μ m/5 μ m, an ITO FeMFET with channel W/L of 50 μ m/15 μ m and FeCAP area of $3 \times 3 \mu$ m², and a micro-LED with

an area of $50 \times 50 \mu\text{m}^2$.

Furthermore, we have updated Supplementary Fig. 4 to include images of the fully assembled flexible display. The complete μ -LED display module, including peripheral signal lines, measures $9.5 \times 8 \text{ mm}^2$ (Fig. R6b). The active pixel area consists of a 4×5 array of 1T-1FeMFET pixel circuits, occupying an area of $820 \times 1070 \mu\text{m}^2$ (Fig. R6c).

Figure R6: (a) Micrograph of 1T-1FeMFET pixel circuit with a bonded μ -LED. (b) Optical image of the complete μ -LED display module with peripheral signal lines. (c) Micrograph of the active μ -LED display array based on the 1T-1FeMFET pixel circuit.

Our modification:

- **Figure 4 in the manuscript:** We have updated Fig. 4a on page 21 to elaborate on the geometric details of the constituent elements in the pixel circuit.
- **Figure 5 in the supplementary:** We have added Figs. R6b and c to Supplementary Figure 5 and updated it accordingly.

Reviewer 2:

This manuscript reports 1T-1FeMFET pixel circuit by incorporating ITO channel and HZO gate stack, suitable for flexible substrates with low processing temperature, high field-effect mobility and low off-currents. The pixel circuit reported by Huang et al. presents competitive statistics with remnant polarization, record memory window, on/off ratio, flexible durability and application to μ LED. We consider this manuscript suitable for publication in Nature Communications, after provided with reply to following comments regarding detailed reasoning for the fabrication and results.

Comment 1:

Details on processing temperature: the authors stated that while HZO-based FeCAPs and FeFETs provide promising platform, the high crystallization temperature limits the practical use on flexible substrates. Thus, in this work ITO channels with lower processing temperatures $< 200 \text{ }^\circ\text{C}$ are used. However, pre-heating of the substrates and processing of HZO layer in the FeMFET presented in this work still requires $400 \text{ }^\circ\text{C}$ of processing. Is $400 \text{ }^\circ\text{C}$ the optimal temperature, and what happens if lower or higher temperature is used to Pr of flexible FeCAP. If the device can endure the temperature and higher temperatures provide higher Pr, that should also be mentioned in the manuscript.

Our response:

We are grateful to the reviewer for this constructive feedback. We firstly explain the motivation for selecting 400°C as the post-metallization annealing (PMA) temperature. The selection of a 400°C annealing temperature was driven by a balance between the maximum tolerable temperature of the flexible substrate and the crystallization requirement of HZO.

1. Specifically, the commercial polyimide (PI) substrate used in this work has a specified maximum tolerable temperature of 450°C. To ensure the mechanical reliability of the substrate, including its flexibility and low residual stress, throughout the full integration flow, a safe thermal margin should be preserved.
2. Furthermore, the polarization of ferroelectric capacitors (FeCAPs) generally exhibits a positive correlation with annealing temperature up to ~ 600°C [15-25]. It is also confirmed in our prior experimental data, as shown in Fig. R7. To enhance both the remnant polarization $2P_r$ of the FeCAPs and the resulting memory window of the ITO FeMFET, it is crucial to apply the highest feasible temperature without compromising the stability of the PI substrate. Thus, the HZO crystallization anneal was conducted at 400°C.

On the other hand, the ITO channel is deposited at room temperature and well activated at a low temperature of 200°C, which is far below the maximum tolerable temperature of the polyimide substrate. More importantly, this step with low temperature has no adverse impact on the FeCAPs fabricated in prior front-end processes.

Figure R7: P-V hysteresis loops expand with increasing PMA temperature.

Our modification:

- **Line 328-331 on page 10 in the manuscript:** We have replaced the sentence “After that, a post metallization annealing (PMA) at 400°C for 2 hours is conducted to crystallize HZO.” with “After that, a post-metallization annealing (PMA) at 400°C for 2 hours is conducted to crystallize the HZO layer, a temperature chosen to balance the maximum tolerable temperature of the flexible polyimide substrate with the need for optimal ferroelectric performance (Supplementary Fig. S19).”.
- **Line 124-127 on pages 4 and 5 in the manuscript:** We have replaced the sentence “As shown in Fig. 2c, the ferroelectric polarization undergoes complete switching at voltages above 3 V. Partial polarization switching can be achieved by reducing the sweep amplitude, as further illustrated in Supplementary Fig. S3.” with “As shown in Fig. 2c, the ferroelectric polarization undergoes complete switching at voltages above 3 V, although this yields insufficient $2P_r$ for FeCAPs after 300°C PMA. Reducing the sweep amplitude enables partial polarization

switching (Supplementary Fig. S3).”.

- **Figure 2 in the manuscript:** We have updated Fig. 2c to more clearly show the positive correlation of ferroelectricity with annealing temperature.
- **Figure 19 in the supplementary:** We have added Fig. R7 as Supplementary Figure 19 to clarify the enhanced ferroelectric performance of FeCAPs achieved with 400°C PMA compared to 300°C PMA.

Comment 2:

Details on channel dimension:

- The authors used channel dimensions of 50/15 μm for ITO FeMFET. While the superiority of 1T-1FeMFET drives from allowing miniaturization of the capacitors, the reduction in channel dimension should also be required. Thus, trend of channel reduction in ITO FeMFET should be tested, as well as mentioned in Table S2 for comparison.

- Also, the details on projected scalability and the projected pixel density of μLED should be provided, and if they can be confirmed experimentally.

Our response:

We appreciate the reviewer’s perspective on this issue. We have investigated the electrical characteristics of the ITO FeMFETs with varying channel dimensions (A_{De}) and identical FeCAP area (A_{Fe}) of $9 \mu\text{m}^2$. As summarized in Fig. R8 and Table R2 (Supplementary Table S2), the M.W. narrows as A_{De} is reduced. As A_{De} is scaled down from $1000 \mu\text{m}^2$ to $150 \mu\text{m}^2$, the M.W. decreases from 9.4 V to 1.2 V, indicating a significant deterioration in ferroelectric behavior (evident from the remaining counterclockwise-CCW hysteresis). Whereas, when A_{De} is further reduced to $25 \mu\text{m}^2$, the ferroelectric behavior vanishes, with the transfer curves exhibiting a conventional clockwise-CW direction. This is attributed to the altered internal voltage division within the MFMIS stack: as A_{De} is scaled down, a smaller fraction of the applied bottom gate voltage (V_{BG}) is dropped across the metal-ferroelectric-metal (MFM) capacitor, which weakens the synergy between ferroelectric switching and charge trapping through two primary mechanisms:

1. **Limited ferroelectric polarization switching:** The lower voltage across the FeCAP prevents full saturation of the P-V hysteresis loop, leading to a minor-loop behavior and thus a diminished polarization contribution to the M.W.
2. **Suppressed charge trapping:** With less voltage dropped across the MFM capacitor, the electric field that drives charge transfer between the bottom gate (BG) and floating gate (FG) is reduced. This attenuates the contribution from charge trapping.

Therefore, the decrease in M.W. with scaled channel dimensions is attributed to the progressive degradation of both ferroelectric and charge-trapping contributions under reduced A_{De} conditions.

Figure R8: (a) Hysteresis transfer curves and (b) extracted M.W. of ITO FeMFET with a FeCAP area fixed at $9 \mu\text{m}^2$ and varying channel dimensions (A_{De}).

The pixel density (PPI) presented in Fig. 4h was calculated under a defined display configuration: a 4-inch diagonal, 16:9 aspect ratio panel. The area of a single pixel was defined as the combined footprint of the driving transistor, the selecting transistor, the storage capacitor (FeCAP in this work), and the μ -LED. PPI was then derived from this effective pixel area and the overall display dimensions. This approach enables a fair and layout-independent comparison across different reported pixel circuits. Using this baseline, the PPI projected for a $10\times$ reduction in critical linewidths reflects the scalable potential of the architecture in advanced fabrication nodes.

Table R2: Benchmarks of M.W. and I_{on}/I_{off} of the ITO FeMFET in this work versus recently reports.

Channel	FE	M.W. (V)	Thickness of FE (nm)	Normalized M.W. (V/nm)	W/L and Area ($\mu\text{m}/\mu\text{m}$, μm^2)	I_{on}/I_{off}	year	Ref.
ITO	HZO	2.78	11.4	0.24	50/5, 250	10^8	2023	8
MoS ₂	HZO	9.5	16	0.59	/	10^7	2023	9
InAs	HZO	1.53	12	0.125	/	10^4	2022	10
IWO	HZO	2	10	0.2	20/0.25, 5	10^5	2022	11
InZnO _x	HZO	3.84	24	0.16	/	10^5	2023	12
IZTO	HZO	2	32	0.063	350/15, 5250	10^5	2021	13
In ₂ O ₃	HZO	2.2	10	0.22	1.5/1.5, 2.25	10^7	2021	14
ITO/IGZO	HZO	2.1	10	0.21	0.5/0.04, 0.02	10^7	2022	15
IWO	HZO	1.45	10	0.145	20/0.25, 5	10^5	2021	16
		9.4		0.78	100/10, 1000	10^9		Target Device
		7.5		0.63	50/15, 750	4×10^8		
ITO	HZO	4.2	12	0.35	100/5, 500	5×10^8	2025	This work
		4		0.33	50/10, 500	10^8		
		2.5		0.21	50/5, 250	10^8		
		1.2		0.1	15/10, 150	10^6		
		0.5 (CW)		0.04 (CW)	5/5, 25	/		

Our modification:

- **Line 166-168 on page 6 in the manuscript:** We have replaced the sentence “Figures 3a and 3b present the transfer at $V_d = 0.1$ V and output at $V_{FG} = 2$ to 5 V characteristics of an ITO FeMFET with $W/L = 50 \mu\text{m}/15 \mu\text{m}$ and a FeCAP size of $3 \times 3 \mu\text{m}^2$.” with “**Figures 3a presents the transfer at $V_d = 0.1$ V and output at $V_{FG} = 2$ to 5 V characteristics of an ITO FeMFET with $W/L = 50 \mu\text{m}/15 \mu\text{m}$ and a FeCAP size of $3 \times 3 \mu\text{m}^2$.**”.
- **Line 177-181 on page 6 in the manuscript:** We have replaced the sentence “Fig. 3c illustrates the hysteresis transfer characteristics of the ITO FeMFET across different ranges of V_{BG} , displaying clear counterclockwise hysteresis loops that expand with increasing V_{BG} sweep amplitude. The V_{th} differences from the forward ($V_{th-pro.}$) and backward ($V_{th-era.}$) sweeps define the memory window (M.W.), which reaches 7.5 V at V_{BG} range of ± 8 V, as depicted in Fig. 3d.” with “**Fig. 3b illustrates the hysteresis transfer characteristics of the ITO FeMFET across different ranges of V_{BG} , displaying clear counterclockwise hysteresis loops that expand with increasing V_{BG} sweep amplitude. The V_{th} differences from the forward ($V_{th-pro.}$) and backward**

(V_{th-era}) sweeps define the memory window (M.W.), which reaches 7.5 V at V_{BG} range of ± 8 V, as depicted in Fig. 3c.”

- **Line 182-183 on page 6 in the manuscript:** We have replaced the sentence “Figure 3e benchmarks the normalized M.W. against on/off ratio” with “Figure 3d benchmarks the normalized M.W. against on/off ratio”.
- **Line 203-208 on page 7 in the manuscript:** We have added the sentence “To further explore the geometry dependence, with a focus on the channel-reduction trend, devices with a fixed A_{Fe} ($9 \mu m^2$) but scaled A_{De} are characterized (Figure 3e and Supplementary Fig. S10). The M.W. narrows with decreasing A_{De} , dropping from 9.4 V at $1000 \mu m^2$ to 1.2 V at $150 \mu m^2$, where a weakened counterclockwise hysteresis remains. Further scaling to $25 \mu m^2$ eliminates the ferroelectric signature, resulting in a purely clockwise hysteresis.”.
- **Figure 10 in the supplementary:** We have added Fig. R8a as Supplementary Figure 10 to detailed exhibit the performance of ITO FeMFET with a FeCAP area fixed at $9 \mu m^2$ and varying A_{De} .
- **Figure 3 in the manuscript:** We have added Fig. R8b as Figure 3d to validate the positive correlation of M.W. with A_{De} .
- **Table 2 in the supplementary:** We have updated Supplementary Table 2 to include the ITO FeMFET with varying A_{De} for comparison.
- **Line 281-287 on page 9 in the manuscript:** We have replaced the sentence “A benchmarking analysis in Fig. 4h compares the pixels per inch (PPI) and dynamic power consumption ($P_{dynamic} = C \cdot V^2 \cdot f$) of this work against previously reported pixel circuits⁴⁹⁻⁵⁵, under standardized conditions with a size of 4 inches, an aspect ratio of 16:9, and a refresh rate of 60 Hz.” with “A benchmarking analysis in Fig. 4h compares the pixels per inch (PPI) and dynamic power consumption ($P_{dynamic} = C \cdot V^2 \cdot f$) of this work against previously reported pixel circuits⁶²⁻⁶⁸. Here, PPI is derived from the effective area of a single pixel, taken as the combined footprint of the driving transistor, selecting transistor, storage capacitor, and the μ -LED, under a defined 4-inch, 16:9, 60 Hz display configuration. This layout-independent approach enables a fair comparison across different pixel circuits.”.

Comment 3:

Details on record normalized memory window: as presented by Table S2, ITO FeMFET presented by the authors demonstrate record M.W. compared to previous studies. While same HZO is used for ferroelectricity, different channel materials are used ranging from oxides to 2D materials. More discussion should be mentioned in the manuscript as to why ITO gives higher M.W. in comparison to other channel materials, what defines the mechanism to the resultant M.W. and why that differs between the chosen materials.

Our response:

We appreciate the reviewer’s insightful comment regarding the memory window (M.W.) comparison across different channel materials. The discussion of how the channel material influences the M.W. is indeed valuable. In our device, the M.W. is primarily governed by the synergistic effect between ferroelectric (FE) switching in the HZO layer and charge trapping at the floating-gate (FG) node. While the choice of channel material—such as ITO, other oxides, or 2D materials—does not dictate the M.W., it can significantly influence carrier mobility and on-state current. The threshold voltage (V_{th}) shift that defines the M.W. originates primarily from the

ferroelectric/dielectric stack and its charge dynamics.

A Deeper Analysis of the Record Memory Window: Synergistic Effect of Ferroelectric (FE) Switching and Charge Trapping

1. Contribution from FE-switching

In a conventional FE-based model, the memory window arises primarily from polarization reversal. Considering the series capacitance of the HZO (C_{HZO}) and Al_2O_3 (C_{De}) layers, the voltage division under Fe-mode operation can be described as:

$$V_{\text{BG}} = V_{\text{HZO}} + V_{\text{De}} + \varphi \quad (1)$$

where V_{HZO} and V_{De} are the voltage drops across the HZO and Al_2O_3 layers, respectively, and φ is the surface potential of the ITO channel. The threshold voltages in the programmed ($V_{\text{th-pro.}}$) and erased ($V_{\text{th-era.}}$) states can be expressed as:

$$V_{\text{th-pro.}} = V_{\text{HZO-pro.}} + V_{\text{DE-pro.}} + \varphi_{\text{pro.}} \quad (2)$$

$$V_{\text{th-era.}} = V_{\text{HZO-era.}} + V_{\text{DE-era.}} + \varphi_{\text{era.}} \quad (3)$$

At threshold, V_{DE} and φ are essentially identical for both states. It is worth noting that the total charges within the ferroelectric HZO layer (Q_{FE}) consist of polarization (P_{HZO}) and non-polarization ($P_{\text{HZO,non}}$). Considering the law of charge conservation:

$$Q_{\varphi} = Q_{\text{FE}} = P_{\text{HZO}} + P_{\text{HZO,non}} = P_{\text{HZO}} + C_{\text{HZO}} \cdot V_{\text{HZO}} \quad (4)$$

where Q_{φ} is the charges in the ITO layer. The memory window due solely to FE switching reduces to:

$$M.W. = \frac{P_{\text{HZO-era.}} - P_{\text{HZO-pro.}}}{C_{\text{HZO}}} \quad (5)$$

Typically, M.W. is bounded by $2V_{\text{C}}$ (twice the coercive voltage of the FE layer). However, our measured M.W. exceeds this limit, indicating an additional contributing mechanism.

2. Role of Charge Trapping

Our ITO FeMFET possesses a high area ratio ($\text{AR} \approx 83$) between the FeCAP ($9 \mu\text{m}^2$) and the ITO TFT ($750 \mu\text{m}^2$). This results in most of the applied V_{BG} dropping across the metal–ferroelectric–metal (MFM) capacitor, with only a small fraction across the ITO TFT. The resulting electric field can promote charge tunneling between the bottom gate (BG) and the floating gate (FG) [26-30], as illustrated in the energy-band diagrams (Figs. R9a and b). Because the Al_2O_3 layer is relatively thick (20 nm) and the voltage across it is small, charge exchange between the BG and FG dominates over that between the ITO channel and FG. During the backward sweep, a positive V_{BG} attracts electrons from the FG toward the BG, inducing a negative shift in V_{th} . Conversely, during the forward sweep, a negative V_{BG} repels electrons from the BG region, causing a positive shift in V_{th} . Both of these charge-trapping processes effectively widen the overall memory window [26-30].

3. Synergistic Effect

The record normalized memory window of 0.63 V/nm (7.5 V absolute) observed in our ITO

FeMFET therefore stems from the synergistic effect between FE-switching and charge trapping. While FE-switching provides the foundational memory effect, charge trapping, favored by the high AR and the resulting voltage division, further expands the window beyond the conventional $2V_C$ limit.

Thus, the memory window (M.W.) of the ITO FeMFET was optimized through a co-design of the ferroelectric capacitor (FeCAP) and the area ratio ($AR = A_{De}/A_{Fe}$). As shown in Figs. R9c, d, and e, the FeCAP performance was first maximized by tuning the HZO thickness and the post-metallization annealing (PMA) temperature. The FeCAPs subjected to PMA at 300°C show inferior ferroelectricity, including relatively low $2P_r$ ($< 15 \mu\text{C}/\text{cm}^2$) and asymmetric coercive voltage. In contrast, those treated at 400°C exhibit markedly enhanced performance, as evidenced by well-expanded P-V hysteresis loops. Among FeCAPs treated with PMA at 400°C, the FeCAPs with a 12-nm-thick HZO layer exhibit the best performance. Additionally, the pristine P-V loops of 8-nm FeCAPs exhibit signs of early breakdown. This is attributed to oxygen vacancies, generated at the metal/HZO interface during PMA, which aggregate near the HZO grain boundaries and create leakage paths. And FeCAPs with a 16-nm HZO layer exhibit compromised performance as a result of the increased fraction of the paraelectric monoclinic phase, which diminishes $2P_r$. Therefore, an HZO thickness of 12 nm combined with a PMA temperature of 400°C is identified as the optimal condition to achieve superior ferroelectric performance in the FeCAPs.

Based on the optimized FeCAP, we tested the hysteresis transfer curves of ITO FeMFETs with a fixed channel dimension (A_{De}) of $750 \mu\text{m}^2$ while varying FeCAP areas (A_{Fe}). As shown in Figs. R9f and g, the M.W. increases monotonically as A_{Fe} is scaled up from 9 to $400 \mu\text{m}^2$ (corresponding to an area ratio decrease from 83 to 1.9). This trend is attributed to the reduced voltage drop across the metal-ferroelectric-metal (MFM) capacitor at larger A_{Fe} . Consequently, the P-V hysteresis loop becomes unsaturated, yielding weaker polarization and less efficient charge transfer between the BG and FG. For the smallest A_{Fe} device, ferroelectric behavior is significantly diminished, owing to minimal polarization reversal and a negligible tunneling current. To achieve the reported M.W. of 7.5 V with a given ITO TFT footprint, however, an impractically large AR would be required, posing scaling challenges in advanced technology nodes. After thorough evaluation, an AR of 83.3 (corresponding to $A_{De} = 750 \mu\text{m}^2$ and $A_{Fe} = 9 \mu\text{m}^2$) was selected as the optimal trade-off between performance and practical integration.

Figure R9: Illustration of the energy band diagrams of the charge trapping in the FG under (a) positive and (b) negative applied voltage. (c)-(e) The performance of FeCAP versus HZO thickness and PMA temperature. (f) Hysteresis transfer curves and (g) extracted memory window (M.W.) of the ITO FeMFET as a function of the AR, with a fixed channel dimension (A_{De}) of $750 \mu\text{m}^2$.

Our modification:

- **Line 189-203 on pages 6 and 7 in the manuscript:** We have replaced the sentence “Record normalized M.W. is attributed to remarkable $2P_r$ and large potential across the HZO layer (V_{HZO}), which is expressed as

$$V_{HZO} = V_{BG} \times \frac{C_{DE}}{C_{DE} + C_{FE}} \quad (1)$$

where C_{FE} and C_{DE} are the capacitance of the HZO and Al_2O_3 layer, respectively²⁷. The memory properties of the ITO FeMFET can further enhance by modulating the ratio of

C_{FE}/C_{DE} .” with “The record M.W. of 0.63 V/nm (7.5 V absolute) stems from the synergistic effect between ferroelectric (FE) switching in the HZO layer and charge trapping at the floating-gate (FG) node. With a channel dimension (A_{DE}) of $750 \mu\text{m}^2$ and a FeCAP area (A_{FE}) of $9 \mu\text{m}^2$, the resulting high area ratio ($AR = A_{DE}/A_{FE}$) of 83 causes the majority of V_{BG} to drop across the HZO layer, thus enabling complete polarization reversal. This FE-switching yields a fundamental M.W. of approximately 3 V, which approaches twice the coercive voltage ($2V_C$) of the HZO layer. Beyond this, the considerable thickness (20 nm) of the Al_2O_3 layer, along with a relatively small voltage dropped across it, results in the charge transfer effect between the BG and FG overwhelming that between the ITO and FG (Supplementary Fig. S8). Subsequently, charge trapping further expands the M.W. beyond the conventional $2V_C$ limit to 7.5 V^{57-61} . Therefore, these two processes act synergistically to achieve the recorded M.W.. Besides, the hysteresis transfer curves of the ITO FeMFETs with a fixed A_{DE} of $750 \mu\text{m}^2$ while varying A_{FE} are investigated (Supplementary Fig. S9), which also serves to validate the aforementioned synergistic effect.”.

- **Line 112-115 on page 4 in the manuscript:** We have replaced the sentence “The high crystallinity of HZO is well defined after the post metallization annealing (PMA) treatment at 400°C .” with “The high crystallinity of the 12-nm-thick HZO layer is well defined after the post metallization annealing (PMA) treatment at 400°C . This temperature, along with a 12-nm HZO thickness, is found to optimize ferroelectric performance after systematic variation of these parameters (Supplementary Fig. S1).”.
- **Figure 8 in the supplementary:** We have added Figs. R9a and b as Supplementary Figure 8 to depict the mechanism of charge trapping.
- **Figure 9 in the supplementary:** We have added Figs. R9c and d as Supplementary Figure 9 to validate the synergistic effect.
- **Figure 1 in the supplementary:** We have added Figs. R9c, d, and e as Supplementary Figure 1 to delineate the optimization details of the M.W. in the ITO FeMFET.
- **Refs. [57], [58], [59], [60], [61] on page 15 in the manuscript:** We have added reference [R26], [R27], [R28], [R29], [R30] as the reference [57], [58], [59], [60], [61].

Comment 4:

Details on the low I_{off} ; line 204 mentioned that I_{off} remains at the measurement limit across all condition, which is at $2 \times 10^{-10} \mu\text{A}/\mu\text{m}$. If this value is extracted due to machine limitation (compliance), is it truly correct value? Please elaborate on this value, and if more accurate data can be extracted by change of device dimensions or alternative measurement.

Our response:

Thank you for this thoughtful and helpful suggestion. The off-state current reported in Figs. 3h and 3i is limited by the measurement setup rather than by the intrinsic performance of the ITO device. Our Lake Shore probe station has a measurement limit of approximately 10 fA (10^{-14} A). For the ITO TFT with $W/L = 50 \mu\text{m}/15 \mu\text{m}$, the normalized I_{off} of $2 \times 10^{-10} \mu\text{A}/\mu\text{m}$ corresponds to this instrument limit. To accurately extract the intrinsic off-state current, we fabricated ITO TFTs with larger W/L ratios. As shown in Fig. R10a, a device with $W/L = 500 \mu\text{m}/15 \mu\text{m}$ exhibits noise around 10 fA, confirming that the measurement is still at the system limit. With an even larger $W/L = 4000 \mu\text{m}/5 \mu\text{m}$ (Fig. R10b), the off-state current is clearly resolved at ~ 100 fA. These results confirm that the intrinsic off-state current of the ITO devices is ultralow and lies significantly below

the detection threshold of our characterization setup.

Figure R10: Transfer curves for the ITO TFTs with W/L of (a) 500 $\mu\text{m}/15 \mu\text{m}$ and (b) 4000 $\mu\text{m}/5 \mu\text{m}$.

Our modification:

- **Line 174-176 on page 6 in the manuscript:** We have added the sentence “Notably, the measured off-state current corresponds to the ~ 10 fA detection limit of the measurement setup (see Supplementary Fig. S7), implying an even lower intrinsic value.”.
- **Figure 7 in the supplementary:** We have added Fig. R10 as Supplementary Figure 7 to demonstrate that the off-state current of the proposed device is limited by the measurement setup.

Comment 5:

Details on Figures

- *Figure 2c. P-V hysteresis loop at various voltage ranges, the voltages are not labeled*
- *Figure S4. Provide scale*
- *Figure 4. Provide emission efficiency.*

Our response:

We thank the reviewer for pointing out this aspect, which we have now addressed. We have revised Fig. 2c in the manuscript, which is shown in Fig. R11a.

Based on the reviewers’ comments, we have updated Supplementary Fig. 4 accordingly. An optical image of the polyimide (PI) film with a scale bar is provided in Fig. R11b. The full display module, including peripheral signal lines, measures $9.5 \times 8 \text{ mm}^2$ (Fig. R11c). The active pixel region consists of a 4×5 array of 1T-1FeMFET pixel circuits and occupies an area of $820 \times 1070 \mu\text{m}^2$ (Fig. R11d).

In addition, the on-wafer external quantum efficiency (EQE) of the μ -LED is presented in Fig. R11e. The EQE peaks at a relatively low current density of 2.5 A/cm^2 , indicating favorable efficiency characteristics under practical operating conditions.

Figure R11: (a) P-V hysteresis loop at various voltage ranges. (b) Optical image of the PI film after laser-lifted-off (LLO). (c) Optical image of the complete μ -LED display module with peripheral signal lines. (d) Micrograph of the active μ -LED display array based on the 1T-1FeMFET pixel circuit. (e) On-wafer external quantum efficiency (EQE) of the μ -LED.

Our modification:

- **Figure 2 in the manuscript:** We have updated Fig. 2c to include the labels of voltage ranges.
- **Line 279-281 on page 9 in the manuscript:** We have added the sentence “Optical image of the complete μ -LED display module with peripheral signal lines and micrograph of the active μ -LED display array are shown in Supplementary Figs. S5b and c.”.
- **Figure 5 in the supplementary:** We have updated Supplementary Figure 5 to illustrate the dimensions of the display array.
- **Line 276-277 on page 9 in the manuscript:** We have added the sentence “The on-wafer external quantum efficiency (EQE) of the μ -LED is presented in Supplementary Fig. S17.”.
- **Figure 17 in the supplementary:** We have added Figs. R11e as Supplementary Figure 17 to supplement the characterization of the μ -LED.

Comment 6:

The authors miss the important work in oxide transistors field reported recently. I recommend that the author refer to the following works: such as Nature 629 (8013), 798-802, Science Advances 11 (43), eadz6914.

Our response:

Thanks for the comment. We have carefully reviewed these two articles along with other recent advances in oxide semiconductors and have incorporated relevant insights into our study. The following discussion and corresponding citations have been added to the revised manuscript to better situate our work within the current field.

From Selenium-alloyed tellurium oxide for amorphous p-channel transistors (Nature, 2024):

This pioneering work demonstrates that incorporating a high-mobility element (Te) within an amorphous oxide matrix (TeO_x) creates a novel p-type semiconductor system. A key design strategy is using selenium alloying to fine-tune the hole concentration and enhance the connectivity of the p-orbital network, which is crucial for achieving high hole mobility ($\sim 15 \text{ cm}^2 \text{ V}^{-1} \text{ s}^{-1}$) and excellent on/off ratios ($10^6 - 10^7$). This paradigm directly informs our material design: their use of Se alloying to precisely fine-tune electrical properties (e.g., hole concentration and orbital connectivity) validates that the performance of oxide-based devices like our FeMFET can be systematically optimized through compositional engineering (e.g., Sn doping, oxygen vacancy control), offering a clear pathway for further enhancing our device's memory window and on/off ratio.

From *Sulfur redox mediator for low-temperature flexible amorphous oxide CMOS electronics* (Sci. Adv., 2025): Wang et al. introduce a groundbreaking sulfur-mediated redox strategy that directly facilitates the in-situ formation of conductive Te-Te networks within an amorphous TeO_2 matrix at an ultralow temperature of 120°C . This approach enables high-performance p-channel TFTs and fully integrated CMOS circuits on flexible substrates. This powerfully reinforces the core premise of our work: their demonstration of high-speed ring oscillators and complex integrated circuits built with this material provides compelling evidence that advanced oxide devices are capable of the high-frequency operation and functional integration that we target, thereby strongly supporting our claim that the 1T-1FeMFET architecture is suitable for high-refresh-rate, high-resolution displays.

Our modification:

- **Line 64-67 on page 3 in the manuscript:** We have replaced the sentence “As a nanoscale channel material, ITO offers a wide bandgap (3.5-4.2 eV), competitive mobility around $30 \text{ cm}^2 \text{ V}^{-1} \text{ s}^{-1}$, and ultra-low off-state current.” with “Within oxide semiconductors, where p-type materials are also advancing^{29,30}, ITO stands out as a nanoscale channel material offering a wide bandgap (3.5-4.2 eV), competitive mobility around \$30 \text{ cm}^2 \text{ V}^{-1} \text{ s}^{-1}\$, and ultra-low off-state current.”.
- **Refs. [29], [30] on page 13 in the manuscript:** We have added the two reports as the reference [29], [30].

Reviewer 3:

This work presents a new configuration integrating FeMFET and ITO channel into a 1T-1FeMFET pixel circuit for flexible active matrix μ -LED display application. The scheme leverages on the merging of a driving transistor and an embedded memory element, allowing to scale down the pixel size. The FeMFET demonstrated record normalized window as well as impressive stability under mechanical stress when bent down to a radius as small as 4 mm. The work further presented extensive characterization of the device and its employment for flexible displays. The following comments would likely improve the quality of the paper:

Comment 1:

Lines 60-65. It is unclear why the high-temperature requirement for HZO deposition/functionalization needs to be mentioned as a disadvantage. It is understood that it is not

replaced by ITO, which is processable at lower temperatures.

Our response:

We sincerely thank the reviewer for this valuable observation. Indeed, higher-temperature processing enhances the thermal stability of the HZO film and reduces sensitivity to subsequent thermal steps, such as the 250°C Al₂O₃ ALD deposition and the 200°C ITO channel activation.

Nevertheless, our commercial polyimide substrate has a specified maximum tolerable temperature of 450°C, beyond which dimensional stability and mechanical integrity may degrade. To ensure reliable substrate performance, we therefore limited the peak process temperature to 400°C. From a practical standpoint, minimizing the overall processing temperature without sacrificing device performance is also critical for cost control and equipment compatibility, as highlighted in numerous reports on low-temperature ferroelectric capacitors [1-21]. Moreover, in a multilayer or 3D integration scheme, a high-temperature HZO step could adversely affect underlying or previously fabricated layers, further underscoring the need for a moderate thermal profile.

Our modification:

- **Line 317-323 on page 10 in the manuscript:** We have replaced the sentence “The PI film possesses extremely low surface roughness and coefficient of thermal expansion of 3 ppm/°C, exhibiting great compatibility with following processes.” with “**The PI film exhibits excellent processing compatibility, characterized by key properties including low surface roughness, a thickness of 38 μm, thermal shrinkage < 0.1% (after 2 hours at 400°C), a coefficient of thermal expansion (CTE) of 3 ppm/°C, and a maximum tolerable temperature of 450°C (specifications provided by the supplier). Given this 450°C limit, the process temperature is set to 400°C to ensure reliable substrate performance.**”.

Comment 2:

What is the thickness of the polyimide film on glass? Polyimide films typically have a maximum workable temperature around 250°C–275°C. Could you please comment on the thermal expansion experienced by the flexible substrate used after performing the steps requiring a high temperature of 400°C?

Our response:

We acknowledge and thank the reviewer for highlighting this point. The polyimide (PI) film used in this work has the following key properties: thickness = 38 μm, thermal shrinkage < 0.1% (after 2 hours at 400°C), maximum tolerable temperature = 450°C, and coefficient of thermal expansion (CTE) = 3 ppm/°C (data provided by Wuxi Alflex Optoelectronic Technology Co., Ltd). Given this low CTE and minimal thermal shrinkage, the dimensional change of the PI substrate after the 400°C post-metallization annealing (PMA, 2 h) is negligible, ensuring the mechanical stability required for subsequent device integration.

Our modification:

- **Line 317-323 on page 10 in the manuscript:** We have replaced the sentence “The PI film possesses extremely low surface roughness and coefficient of thermal expansion of 3 ppm/°C, exhibiting great compatibility with following processes.” with “**The PI film exhibits excellent processing compatibility, characterized by key properties including low surface roughness, a**

thickness of 38 μm , thermal shrinkage $< 0.1\%$ (after 2 hours at 400°C), a coefficient of thermal expansion (CTE) of 3 $\text{ppm}/^\circ\text{C}$, and a maximum tolerable temperature of 450°C (specifications provided by the supplier). Given this 450°C limit, the process temperature is set to 400°C to ensure reliable substrate performance.”.

Comment 3:

What are the critical interfaces/layers during bending? For the mechanical stability test performed by bending, please report the estimated induced strain. How does the crystallinity of HZO affect or support the mechanical performance of the circuit?

Our response:

Thank you for this thoughtful and helpful suggestion. Given that the thickness of the polyimide substrate ($t_{PI} = 38 \mu\text{m}$) is greater than the total thickness of the buffer and transistor stack ($< 1 \mu\text{m}$), and that the bending radius (r_{bending}) is much larger than t_{PI} , the bending strain ϵ experienced by the device can be approximated using the classic beam-bending formula [37,38]:

$$\epsilon = \frac{t_{PI}}{2 \cdot r_{\text{bending}}} \quad (1)$$

The calculated ϵ values are 0.17%, 0.27%, 0.38%, and 0.48% for $r_{\text{bending}} = 11 \text{ mm}$, 7 mm, 5 mm, and 4 mm, respectively. Due to its dominant thickness, the polyimide (PI) film governs the overall strain imposed on the stacked devices during bending.

Despite the high local strain under tight bending, the FeCAP itself is extremely small ($3 \times 3 \mu\text{m}^2$) and therefore undergoes negligible deformation. Consequently, its polarization characteristics and the drain current (I_d) of the ITO FeMFET remain unaffected, as confirmed by the bending tests.

Our modification:

- **Line 139-143 on page 5 in the manuscript:** We have replaced the sentence “For static bending test, samples are mounted on arched fixtures with bending radius (r_{bending}) of 11 mm, 7 mm, and 5 mm. Cyclic bending tests defined one bending cycle as a transition from flat to 4-mm r_{bending} and back, repeated for 10^5 cycles.” with “For static bending test, samples are mounted on arched fixtures with bending radius (r_{bending}) of 11 mm, 7 mm, and 5 mm, which correspond to calculated bending strains (ϵ) of 0.17%, 0.27%, and 0.38%, respectively. Cyclic bending tests defined one bending cycle as a transition from flat to 4-mm r_{bending} (ϵ of 0.48%) and back, repeated for 10^5 cycles.”.

Comment 4:

Please add a comment on the presence of hump in the transfer characteristics Fig. 3a and 3g.

Our response:

We sincerely thank the reviewer for this valuable observation. The hump observed in the transfer curves (Fig. 3) originates from the SU-8 passivation layer used in the final process step. In our processing conditions, the SU-8 layer exhibits a minor affinity for water-vapor absorption from ambient air. The absorbed water acts as a donor via hydrogen-related species, releasing electrons into the ITO channel. These electrons form a parasitic conduction path at the back interface of the ITO film, resulting in the characteristic hump in the subthreshold region [39-44].

Our modification:

- **Line 170-172 on page 6 in the manuscript:** We have added the sentence “Due to ambient-induced parasitic conduction path at the back channel, a minor hump is observable⁴⁴⁻⁴⁹.”.
- **Refs. [44], [45], [46], [47], [48], [49] on page 13 in the manuscript:** We have added reference [R39], [R40], [R41], [R42], [R43], [R44] as the reference [44], [45], [46], [47], [48], [49].

Comment 5:

In Fig. 4d, there is more than 4 orders of magnitude increase in the off-current of the transistor when the selecting transistor is on. What does this increase imply?

Our response:

The authors appreciate the reviewer’s insightful comment. The variation in off-current (I_{off}) between the on/off states of the selecting transistor, along with the rise in I_{off} and drop in I_{on} during retention testing (Fig. 4d), can be traced to the synergistic effect between ferroelectric (FE) switching in the HZO layer and charge trapping at the floating-gate node. A mechanistic analysis of these two contributions is provided in the following discussion.

- (i) **FE-switching:** Ferroelectric behavior arises from the uniform polarization switching and perfect alignment of ferroelectric domains. This aligned configuration represents a metastable state that evolves spontaneously toward a more stable, lower-energy configuration under various non-ideal driving mechanisms, including depolarization fields, defect migration, and thermal activation [16, 23, 45-48]. As a result, the progressive reversal of a portion of the ferroelectric domains modifies the conductivity and carrier density in the ITO channel, leading to increased I_{off} and decreased I_{on} .
- (ii) **Charge trapping:** According to Fig. R12b, the applied erasing pulse drives electrons to tunnel from the BG through the HZO layer and become trapped at the FG node, which effectively contributes to M.W. enlargement. When the Selecting-T is turned on with V_{DATA} held at 0 V, this ground potential is conveyed to the BG of the Driving-T, thereby accelerating the de-trapping of the electrons trapped at the FG node. Simultaneously, the activated Selecting-T provides a discharge path for the de-trapping charges.

Together, these two processes act synergistically to produce the pronounced increase in I_{off} observed during retention.

Given that the phenomenon in Fig. 4d originates from the same mechanism as the record memory window, a detailed analysis of the synergistic effect between FE-switching and charge trapping is provided below.

1. Contribution from FE-switching

In a conventional FE-based model, the memory window arises primarily from polarization reversal. Considering the series capacitance of the HZO (C_{HZO}) and Al_2O_3 (C_{De}) layers, the voltage division under Fe-mode operation can be described as:

$$V_{\text{BG}} = V_{\text{HZO}} + V_{\text{De}} + \phi \quad (1)$$

where V_{HZO} and V_{De} are the voltage drops across the HZO and Al_2O_3 layers, respectively, and ϕ is the surface potential of the ITO channel. The threshold voltages in the programmed ($V_{\text{th-pro}}$) and

erased ($V_{th-era.}$) states can be expressed as:

$$V_{th-pro.} = V_{HZO-pro.} + V_{DE-pro.} + \varphi_{pro.} \quad (2)$$

$$V_{th-era.} = V_{HZO-era.} + V_{DE-era.} + \varphi_{era.} \quad (3)$$

At threshold, V_{DE} and φ are essentially identical for both states. It is worth noting that the total charges within the ferroelectric HZO layer (Q_{FE}) consist of polarization (P_{HZO}) and non-polarization ($P_{HZO,non}$). Considering the law of charge conservation:

$$Q_{\varphi} = Q_{FE} = P_{HZO} + P_{HZO,non} = P_{HZO} + C_{HZO} \cdot V_{HZO} \quad (4)$$

where Q_{φ} is the charges in the ITO layer. The memory window due solely to FE switching reduces to:

$$M.W. = \frac{P_{HZO-era.} - P_{HZO-pro.}}{C_{HZO}} \quad (5)$$

Typically, M.W. is bounded by $2V_C$ (twice the coercive voltage of the FE layer). However, our measured M.W. exceeds this limit, indicating an additional contributing mechanism.

2. Role of Charge Trapping

Our ITO FeMFET possesses a high area ratio ($AR \approx 83$) between the FeCAP ($9 \mu m^2$) and the ITO TFT ($750 \mu m^2$). This results in most of the applied V_{BG} dropping across the metal-ferroelectric-metal (MFM) capacitor, with only a small fraction across the ITO TFT. The resulting electric field can promote charge tunneling between the bottom gate (BG) and the floating gate (FG) [26-30], as illustrated in the energy-band diagrams (Figs. R12a and b). Because the Al_2O_3 layer is relatively thick (20 nm) and the voltage across it is small, charge exchange between the BG and FG dominates over that between the ITO channel and FG. During the backward sweep, a positive V_{BG} attracts electrons from the FG toward the BG, inducing a negative shift in V_{th} . Conversely, during the forward sweep, a negative V_{BG} repels electrons from the BG region, causing a positive shift in V_{th} . Both of these charge-trapping processes effectively widen the overall memory window [26-30].

3. Synergistic Effect

The record normalized memory window of 0.63 V/nm (7.5 V absolute) observed in our ITO FeMFET therefore stems from the synergistic effect between FE-switching and charge trapping. While FE-switching provides the foundational memory effect, charge trapping, favored by the high AR and the resulting voltage division, further expands the window beyond the conventional $2V_C$ limit.

Figure R12: Illustration of the energy band diagrams of the charge trapping in the FG under (a) positive and (b) negative applied voltage.

Our modification:

- **Line 189-203 on pages 6 and 7 in the manuscript:** We have replaced the sentence “Record normalized M.W. is attributed to remarkable $2P_T$ and large potential across the HZO layer (V_{HZO}), which is expressed as

$$V_{HZO} = V_{BG} \times \frac{C_{DE}}{C_{DE} + C_{FE}} \quad (1)$$

where C_{FE} and C_{DE} are the capacitance of the HZO and Al₂O₃ layer, respectively²⁷. The memory properties of the ITO FeMFET can further enhance by modulating the ratio of C_{FE}/C_{DE} .” with “The record M.W. of 0.63 V/nm (7.5 V absolute) stems from the synergistic effect between ferroelectric (FE) switching in the HZO layer and charge trapping at the floating-gate (FG) node. With a channel dimension (A_{De}) of 750 μm^2 and a FeCAP area (A_{Fe}) of 9 μm^2 , the resulting high area ratio ($AR = A_{De}/A_{Fe}$) of 83 causes the majority of V_{BG} to drop across the HZO layer, thus enabling complete polarization reversal. This FE-switching yields a fundamental M.W. of approximately 3 V, which approaches twice the coercive voltage ($2V_C$) of the HZO layer. Beyond this, the considerable thickness (20 nm) of the Al₂O₃ layer, along with a relatively small voltage dropped across it, results in the charge transfer effect between the BG and FG overwhelming that between the ITO and FG (Supplementary Fig. S8). Subsequently, charge trapping further expands the M.W. beyond the conventional $2V_C$ limit to 7.5 V⁵⁷⁻⁶¹. Therefore, these two processes act synergistically to achieve the recorded M.W.. Besides, the hysteresis transfer curves of the ITO FeMFETs with a fixed A_{De} of 750 μm^2 while varying A_{Fe} are investigated (Supplementary Fig. S9), which also serves to validate the aforementioned synergistic effect.”.

- **Line 267-269 on page 9 in the manuscript:** We have added the sentence “A larger increase in I_{off} is observed, driven by the ground potential at V_{DATA} accelerating charge de-trapping, coupled with the activation of a discharge path.”.
- **Figure 8 in the supplementary:** We have added Figs. R12a and b as Supplementary Figure 8 to depict the mechanism of charge trapping.

- **Refs. [57], [58], [59], [60], [61] on page 15 in the manuscript:** We have added reference [R26], [R27], [R28], [R29], [R30] as the reference [57], [58], [59], [60], [61].

Reference

- [R1] Chang, T.-J. *et al.* Atomic tailoring of low-thermal-budget and nearly wake-up-free ferroelectric $\text{Hf}_{0.5}\text{Zr}_{0.5}\text{O}_2$ nanoscale thin films by atomic layer annealing. *Applied Surface Science* **591** (2022).
- [R2] Chen, Y. *et al.* Flexible $\text{Hf}_{0.5}\text{Zr}_{0.5}\text{O}_2$ ferroelectric thin films on polyimide with improved ferroelectricity and high flexibility. *Nano Research* **15**, 2913-2918 (2021).
- [R3] Jeong, S. *et al.* All-Sputter-Deposited $\text{Hf}_{0.5}\text{Zr}_{0.5}\text{O}_2$ Double-Gate Ferroelectric Thin-Film Transistor With Amorphous Indium–Gallium–Zinc Oxide Channel. *IEEE Electron Device Letters* **44**, 749-752 (2023).
- [R4] Ju, C. *et al.* Improved ferroelectric properties of CMOS back-end-of-line compatible $\text{Hf}_{0.5}\text{Zr}_{0.5}\text{O}_2$ thin films by introducing dielectric layers. *Journal of Materiomics* **10**, 277-284 (2024).
- [R5] Kim, S. J. *et al.* Low-thermal-budget (300 °C) ferroelectric TiN/ $\text{Hf}_{0.5}\text{Zr}_{0.5}\text{O}_2$ /TiN capacitors realized using high-pressure annealing. *Applied Physics Letters* **119** (2021).
- [R6] Kim, S. J. *et al.* A Comprehensive Study on the Effect of TiN Top and Bottom Electrodes on Atomic Layer Deposited Ferroelectric $\text{Hf}_{0.5}\text{Zr}_{0.5}\text{O}_2$ Thin Films. *Materials* **13** (2020).
- [R7] Kim, S. J. *et al.* Effect of film thickness on the ferroelectric and dielectric properties of low-temperature (400 °C) $\text{Hf}_{0.5}\text{Zr}_{0.5}\text{O}_2$ films. *Applied Physics Letters* **112** (2018).
- [R8] Liu, Y. *et al.* Zirconium-Rich Strategy in Ultrathin $\text{Hf}_{0.5}\text{Zr}_{0.5}\text{O}_2$ toward Back-End-of-Line-Compatible Ferroelectric Random Access Memory. *Advanced Science* **12** (2025).
- [R9] Lyu, S. *et al.* Achieving High-Endurance Ferroelectricity in $\text{Hf}_{0.5}\text{Zr}_{0.5}\text{O}_2$ Thin Films on Ge Substrate Through ZrO_2 Interfacial Layer Induced Low-Temperature Annealing. *IEEE Electron Device Letters* **45**, 348-351 (2024).
- [R10] O'Connor, É. *et al.* Stabilization of ferroelectric $\text{Hf}_x\text{Zr}_{1-x}\text{O}_2$ films using a millisecond flash lamp annealing technique. *APL Materials* **6** (2018).
- [R11] Onaya, T. *et al.* Wake-up-free properties and high fatigue resistance of $\text{Hf}_x\text{Zr}_{1-x}\text{O}_2$ -based metal–ferroelectric–semiconductor using top ZrO_2 nucleation layer at low thermal budget (300 °C). *APL Materials* **10** (2022).
- [R12] Tong, K. *et al.* Enhancement of Ferroelectricity in $\text{Hf}_{0.5}\text{Zr}_{0.5}\text{O}_2$ via Pre-Crystallization and Interface Engineering at Ultra-Low Temperature (300 °C) Annealing. *IEEE Electron Device Letters* **46**, 928-931 (2025).
- [R13] Wang, C.-I. *et al.* Atomic layer deposited TiN capping layer for sub-10 nm ferroelectric $\text{Hf}_{0.5}\text{Zr}_{0.5}\text{O}_2$ with large remnant polarization and low thermal budget. *Applied Surface Science* **570** (2021).
- [R14] Yamada, H. *et al.* Energy-Efficient Annealing Process of Ferroelectric $\text{Hf}_{0.5}\text{Zr}_{0.5}\text{O}_2$ Capacitor Using Ultraviolet-LED for Green Manufacturing. *IEEE Journal of the Electron Devices Society* **12**, 195-200 (2024).
- [R15] Joh, H. *et al.* Flexible Ferroelectric Hafnia-Based Synaptic Transistor by Focused-Microwave Annealing. *ACS Applied Materials & Interfaces* **14**, 1326-1333 (2021).
- [R16] Jung, Y. C. *et al.* Robust low-temperature (350 °C) ferroelectric $\text{Hf}_{0.5}\text{Zr}_{0.5}\text{O}_2$ fabricated using anhydrous H_2O_2 as the ALD oxidant. *Applied Physics Letters* **121** (2022).
- [R17] Kang, J. *et al.* Thermal budget study to simultaneously achieve low-temperature (< 400 °C) process and high endurance of ferroelectric $\text{Hf}_{0.5}\text{Zr}_{0.5}\text{O}_2$ thin films. *Applied Physics Letters* **126** (2025).

- [R18] Lee, J. *et al.* Ferroelectricity of $\text{Hf}_{0.5}\text{Zr}_{0.5}\text{O}_2$ Thin Film Induced at 350 °C by Thermally Accelerated Nucleation During Atomic Layer Deposition. *IEEE Transactions on Electron Devices* **71**, 2690-2695 (2024).
- [R19] Lee, Y. *et al.* Boosting non-volatile memory performance with exhalative annealing: A novel approach to low-temperature crystallization of hafnia based ferroelectric. *Materials Today Nano* **28** (2024).
- [R20] Persson, A. E. O. *et al.* Reduced annealing temperature for ferroelectric HZO on InAs with enhanced polarization. *Applied Physics Letters* **116** (2020).
- [R21] Wang, X., Mikolajick, T. & Grube, M. Sputtered Ferroelectric Hafnium–Zirconium Oxide with High Remanent Polarization after Back-End-of-Line Compatible Annealing. *ACS Applied Electronic Materials* **4**, 6142-6148 (2022).
- [R22] Hasan, M. M. *et al.* High performance, amorphous InGaZnO thin-film transistors with ferroelectric ZrO_2 gate insulator by one step annealing. *Applied Surface Science* **611** (2023).
- [R23] Jung, M. *et al.* High Pressure Microwave Annealing Effect on Electrical Properties of $\text{Hf}_x\text{Zr}_{1-x}\text{O}$ Films near Morphotropic Phase Boundary. *ACS Applied Electronic Materials* **5**, 4826-4835 (2023).
- [R24] Lin, Y.-C., McGuire, F. & Franklin, A. D. Realizing ferroelectric $\text{Hf}_{0.5}\text{Zr}_{0.5}\text{O}_2$ with elemental capping layers. *Journal of Vacuum Science & Technology B, Nanotechnology and Microelectronics: Materials, Processing, Measurement, and Phenomena* **36** (2018).
- [R25] Mohit, Haga, K.-i. & Tokumitsu, E. Electrical properties of yttrium-doped hafnium-zirconium dioxide thin films prepared by solution process for ferroelectric gate insulator TFT application. *Japanese Journal of Applied Physics* **59** (2020).
- [R26] Myeong, I. *et al.* Strategies for a Wide Memory Window of Ferroelectric FET for Multilevel Ferroelectric VNAND Operation. *IEEE Electron Device Letters* **45**, 1185-1188 (2024).
- [R27] Wang, X. *et al.* Comprehensive Experiments and Modeling Applicable for Ferroelectric Transistors With an MFMIS Structure and a Wide Range of Area Ratios: Unveiling the Operation Mechanisms. *IEEE Transactions on Electron Devices* **71**, 5332-5338 (2024).
- [R28] Wang, X. *et al.* Deep insights into the Interplay of Polarization Switching, Charge Trapping, and Soft Breakdown in Metal-Ferroelectric-Metal-Insulator-Semiconductor Structure: Experiment and Modeling. *2022 International Electron Devices Meeting (IEDM)* (2022).
- [R29] Wang, X. *et al.* Unveiling the Intricate Dynamic Characteristics of FeFETs with a MFMIS Structure: Experiment and Modeling. *2024 IEEE International Electron Devices Meeting (IEDM)* (2024).
- [R30] Yan, M.-H. *et al.* BEOL-Compatible Multiple Metal-Ferroelectric-Metal (m-MFM) FETs Designed for Low Voltage (2.5 V), High Density, and Excellent Reliability. *2020 IEEE International Electron Devices Meeting (IEDM)* (2020).
- [R31] Huang, Z. R. *et al.* Enhancing FeFET Performance Through H_2 Plasma Treatment: Improving Stability, Conductance, and Hamming Distances in FeCAM Designs. *2024 IEEE International Electron Devices Meeting (IEDM)* (2024).
- [R32] Chen, Z. *et al.* Novel Design Strategy for High-Endurance ($> 10^{10}$) and Fast-Erase Oxide-Semiconductor Channel FeFET. *2024 IEEE International Electron Devices Meeting (IEDM)* (2024).

- [R33] Duan, J. *et al.* Variation Tolerant and Energy-Efficient Charge Domain Compute-in-Memory Array with Binary and Multi-Level Cell Ferroelectric FET. *2024 IEEE International Electron Devices Meeting (IEDM)* (2024).
- [R34] Joh, H. *et al.* Oxide Channel Ferroelectric NAND Device with Source-Tied Covering Metal Structure: Wide Memory Window (14.3 V), Reliable Retention (> 10 Years) and Disturbance Immunity ($\Delta V_{th} \leq 0.1$ V) for QLC Operation. *2024 IEEE International Electron Devices Meeting (IEDM)* (2024).
- [R35] Kim, G. *et al.* Unveiling the Origin of Disturbance in FeFET and the Potential of Multifunctional TiO₂ as a Breakthrough for Disturb-Free 3D NAND Cell: Experimental and Modeling. *2024 IEEE International Electron Devices Meeting (IEDM)* (2024).
- [R36] Kuk, S.-H. *et al.* Superior QLC Retention (10 Years, 85 °C) and Record Memory Window (12.2 V) by Gate Stack Engineering in Ferroelectric FET: from “MIFIS” to “MIKFIS”. *2024 IEEE International Electron Devices Meeting (IEDM)* (2024).
- [R37] Han, K.-L., Jeong, H.-J., Kim, B.-S., Lee, W.-B. & Park, J.-S. Recent review on improving mechanical durability for flexible oxide thin film transistors. *Journal of Physics D: Applied Physics* **52** (2019).
- [R38] Lai, H.-C., Pei, Z., Jian, J.-R. & Tzeng, B.-J. Alumina nanoparticle/polymer nanocomposite dielectric for flexible amorphous indium-gallium-zinc oxide thin film transistors on plastic substrate with superior stability. *Applied Physics Letters* **105** (2014).
- [R39] Horng Nan, C., Chung Len, L. & Tan Fu, L. The effects of fluorine passivation on polysilicon thin-film transistors. *IEEE Transactions on Electron Devices* **41**, 698-702 (1994).
- [R40] Kim, J. *et al.* A study on H₂ plasma treatment effect on a-IGZO thin film transistor. *Journal of Materials Research* **27**, 2318-2325 (2012).
- [R41] Tari, A. & Wong, W. S. Effect of dual-dielectric hydrogen-diffusion barrier layers on the performance of low-temperature processed transparent InGaZnO thin-film transistors. *Applied Physics Letters* **112** (2018).
- [R42] Toda, T., Deapeng, W., Jingxin, J., Mai Phi, H. & Furuta, M. Quantitative Analysis of the Effect of Hydrogen Diffusion from Silicon Oxide Etch-Stopper Layer into Amorphous In–Ga–Zn–O on Thin-Film Transistor. *IEEE Transactions on Electron Devices* **61**, 3762-3767 (2014).
- [R43] Yu, Z. *et al.* A High Voltage Gain Inverter Integrated With Enhancement- and Depletion-Mode a-InGaZnO Thin-Film Transistors. *IEEE Transactions on Electron Devices* **70**, 4963-4967 (2023).
- [R44] Zhang, Y. *et al.* Effects of Passivation Layers on the Characteristics and Stability of Indium–Gallium–Zinc Oxide Thin-Film Transistors. *IEEE Transactions on Electron Devices* **72**, 4150-4155 (2025).
- [R45] Ashikaga, K. & Ito, T. Analysis of memory retention characteristics of ferroelectric field effect transistors using a simple metal–ferroelectric–metal–insulator–semiconductor structure. *Journal of Applied Physics* **85**, 7471-7476 (1999).
- [R46] Qin, Y. *et al.* Clarifying the Role of Ferroelectric in Expanding the Memory Window of Ferroelectric FETs with Gate-Side Injection: Isolating Contributions from Polarization and Charge Trapping. *2024 IEEE International Electron Devices Meeting (IEDM)* (2024).
- [R47] Yoon, S.-J. *et al.* Improvement in Long-Term and High-Temperature Retention Stability of Ferroelectric Field-Effect Memory Transistors With Metal–Ferroelectric–Metal–Insulator–

Semiconductor Gate-Stacks Using Al-Doped HfO₂ Thin Films. *IEEE Transactions on Electron Devices* **67**, 499-504 (2020).

[R48] Zheng, Y. *et al.* Polarization Degradation and Recovery Strategies of Hafnia-Based Ferroelectric Capacitors After Thermal Budget in Back-End of Line Process. *2024 IEEE International Electron Devices Meeting (IEDM)* (2024).